# The Reform of School Catering in Hungary: Anatomy of a Health-Education Attempt

**DOI:** 10.3390/nu11040716

**Published:** 2019-03-27

**Authors:** Anna Kiss, József Popp, Judit Oláh, Zoltán Lakner

**Affiliations:** 1Faculty of Food Science, Szent István University, Budapest 1118, Hungary; kiss.anna891@gmail.com (A.K.); lakner.zoltan@etk.szie.hu (Z.L.); 2Institute of Sectoral Economics and Methodology, Faculty of Economics and Business, University of Debrecen, Debrecen 4032, Hungary; popp.jozsef@econ.unideb.hu; 3Institute of Applied Informatics and Logistics, Faculty of Economics and Business, University of Debrecen, Debrecen 4032, Hungary

**Keywords:** institutional economics, MACTOR model, nutrition policy, prevention, semi-quantitative methods, strategic analysis

## Abstract

School lunch nutrition standards are an important carrier of messages on healthy eating and an efficient way of changing the nutritional behaviour of new generations. Many countries in Europe have a compulsory system of school meals; the Hungarian government also wanted to take action in order to improve the nutrition requirements of the school catering service. The Hungarian Ministry of Human Resources established some limits in the school catering system. However, increasing public pressure forced the legislating organ to considerably modify this regulation. The aim of this study is to analyse the causes of this failure, based on a conceptual framework of institutional economics and a strategic modelling of different institutes by examining the results of 72 interviews (33 experts, 26 parents and 13 teachers) conducted with representatives of different stakeholders. The results highlight the lack of preparation for the introduction of the new regulatory framework, as well as the inefficient communication between the different stakeholders. In order to support children in eating healthfully, a complex nutrition education program and continuous dialogue is needed between teachers, parents, catering staff and the government.

## 1. Introduction

Parallel with increasing awareness of the adverse effects of non-communicable diseases, e.g., the effects of obesity and obesity-related diseases on the health condition and mortality of the population [1,2,3], the inappropriate diets and eating habits of children and adolescents is a much-debated problem all over Europe [4,5,6]. Growing anxiety concerning the proliferation of the unhealthy eating habits of new generations [7], as well as the increasing number of overweight and obese children [8,9,10], has increased attention towards different methods of influencing children‘s and adolescents’ health behaviour [11,12]. This is an especially important problem in Hungary where the obesity rate is high. The overall prevalence of overweight and obesity among children and adolescents is 40% and 32%, respectively, and in women overweight and obesity are both at 32% [13]. This causes a considerable burden for the social security and health care system [13,14] according to the categorisation of nutritional status of people based on the body mass index (BMI) [15].

Obesity has been described by the World Health Organization (WHO) as “a global epidemic” due to its high prevalence. It is well known that there are different methods for evaluation of nutritional status of children [16]. Hungary has applied different categorization of nutritional status of children and youth, for example percentiles [17], waist circumference [18] or BMI [19]. All of the researchers have agreed that increasing of obesity is a continuous trend in Hungary. The school catering system (SCS) has a predominant place in the formation of nutrition behaviour [20,21,22], even if the efficiency of the SCS in the prevention of obesity has not been proven satisfactorily by rigorous, long-range studies [23,24]. School lunch nutrition standards are the basis for improving the nutritional intake of all schoolchildren. All member states of the EU have policies to help schools to provide nutritionally balanced meals. The state of school feeding is considered as a question of strategic importance all over the world [25]. All member states of the EU have policies to help schools to provide nutritionally balanced meals [26]. Based on the EU survey [26] and the cluster analysis [27], we have constructed a general synoptic table (Appendix A). It is obvious that there are considerably differences in the school feeding policy of various EU member states (Figure 1). The Hungarian regulation is rather similar to the Spanish and Danish systems.

It can be stated that in developed countries, there is an increasing tendency to apply the SCS not just as a part of school logistics and the social system, but also as means of health and family-life education. That is why the importance of SCS has increased rapidly. The quality of the Hungarian SCS has been a highly debated issue for generations [29]. There are different business models: the majority of kitchens are owned by local authorities; however, several kitchens are run by catering service provider enterprises. Hungary’s SCS is a subsidised system where meals are offered by a reduced price, and in some cases (based on social position) the school meal is free. The Hungarian National Survey of SCS (called as Canteen Panorama) is a regular report of the National Institute of Pharmacy and Nutrition (former Hungarian Office of Food and Nutrition) [30,31,32]. The monitoring and enforcement of the school meal standards is a mandatory regular activity of the National Food Safety Office (NFSO) with government functions. The essential results of the last three surveys are summarised in Table 1. Vending machines and snack bars are not considered as a part of SCS, but their product portfolio is strictly regulated.

The Hungarian SCS is an evolving system, which is characterised by considerable backwardness and delayed development, compared to Western-European school boarding solutions. The national averages hide important regional differences. According to the Canteen Panoramas [30,31,32], the school snack bars and vending machines are significant competitors for the SCS, and have an improving product portfolio. At the same time, the SCS is not capable of meeting changing demands. A good indicator of this is the rapidly decreasing take up rate as children’s age increases. Another indicator of problems is the fact that according to the school canteen survey Bakacs et al. [32], 85% of parents prepared some kind of prepacked food for their children in 2013. Prepacked food was mainly sandwiches, with vegetables (41%) or without vegetables (37%), and refreshing soft drinks (50%).

Generally speaking, the Hungarian SCS faces substantial long-standing, unsolved problems, which can be attributed to the lack of monetary resources and the lack of attention from responsible organs. This motivated the government to take action to update the nutrition requirements of the public catering service, including school meals, in 2011. In 2011, the Office of the Hungarian Chief Medical Officer issued the ‘Recommendation for Public Caterers’ with nutritional standards [33]. This recommendation provided a checklist enabling to monitor the adherence to the recommendations. This document was the basis of the 37/2014 decree of Ministry of Human Capacities (MHC) on public catering [34].

The aim of the school meal provision was to reduce the prevalence of obesity and non-communicable diseases (NCDs) among Hungarian children and adolescents, as well as promote healthier environments, especially in schools. The most important elements of the decree are summarised in Table 2. This rather ambitious regulation has tried to increase the fruit, vegetable, cereals and milk consumption and decrease the consumption of fat, sugar and salt. When comparing the content of the Decree with dietary guidelines of other member states of the EU, our nation’s dietary guidelines seem to be in line with the nutrition policy of most EU member states. However, the public acceptance of this new regulation has been mixed, and mainly negative. Overall take up rates were generally low, according to the comments of school children. Pupils refused the dishes that conformed to the requirements and both children and their parents rebelled against the rules. In 2015, the Hungarian Association of Dietitians carried out a survey among dietitian food-service managers about the practical feasibility of Hungarian Regulation No. 37/2014 on nutrition requirements in the provision of public food services. Of the 56 food-service managers interviewed, 19 represented child nutrition institutions. Since the introduction of the regulation in 36 of 56 institutions interviewed, satisfaction with nutrition care had decreased. In 13 cases, the rate of dissatisfaction was 30% or more, and the amount of daily food waste increased significantly. The majority of catering service providers (62%) requested some alterations to the regulations because the prescribed composition of the food was not in line with children’s demands. The greatest cause of dissatisfaction among parents and children derived from the control of salt content, and the attempt to provide the prescribed quantity of dairy products and added sugars [35]. To date, there are no representative, academically well-founded empirical studies on pupils’ consumption of the new school meals [36]. That is why, in 2016, the regulatory framework was changed [37]. The new decree significantly modified the prescriptions. The most important features of the original and the modified decree and the energy limits of school feeding are summarised in Table 2 and Table 3.

The Hungarian SCS is a highly complex, dynamic system. The last Decree on Public Catering has yielded important results: the quality of meals has been increasing in the last few years [40]; however, numerous unresolved problems have remained. The new regulatory framework of the SCS is more liberal, although in numerous points it represents a step back from the goals of the former Decree. The aim of this study is to uncover the causes of this phenomenon. In a wider context, our goal is to understand the justification of the low acceptance of the SCS towards a regulatory attempt to manage a long-standing, generally accepted problem, namely childhood obesity. What is the main reason that the regulation could not be put into practice? How can we explain that the take up rate of the SCS has not increased considerably since the introduction of the new decree? To achieve this goal, the authors have applied an innovative, semi-quantitative method for the analysis of social bargaining.

## 2. Materials and Methods

### 2.1. The Methodological Framework of the Research

The fundamental theoretical paradigms of the analysis were institutional economic theory [41,42], principle-agent theory [43] and the concept of strategic planning [44]. According to the basic theory of the so-called “French school of strategy”, the different social systems can be considered as a playground in which different groups of participants (the actors) take part with the purpose of making their specific interests prevail. In the opinion of [45], if one can adequately simplify the actors and the most characteristic features of their systems of interests and strategies, then it is possible to analyse the chances of different actors realizing their goals. The method of the systematic analysis of social bargaining can be described by using the MACTOR model. One of the key concepts of the model is that actors may influence other actors in terms of their potential to put pressure on other actors directly or indirectly in order to affect their behaviour. The influence of one actor (A) on another actor (C) is the sum of the direct and indirect influences of actor A on actor C.

Based on unstructured interviews, the key actors of the catering system were determined. In the next phase the intensity of mutual direct influences was characterized using a rectangular matrix offering a good overview of the MACTOR method. The cells of the matrix, by definition, reflect the intensity of the influence of any actor in a row on any actor in a column. The intensity of the direct influence by one actor on another was measured on a 0–4 scale ranging from no influence to absolute influence.

The importance of different goals from the point of view of each actor was expressed by the Matrix of Actor-Objective. This was the so-called 1MAO matrix. Each cell of the matrix contained the attitude of a given actor towards a given goal in the form of a positive, 0 or negative sign. In the second phase the 2MAO matrix is determined, which contains the intensity of these attitudes determined from the point of view of different actors and quantified on a −4 …+4 scale, where −4 denotes the high importance and total negation of the given goal, and +4 denotes the high importance and total support.

The mathematical methodology of the MACTOR method is presented in the literature [46].

### 2.2. Setting Up the Input Data System

The data collection for the analysis was a multiphase process (Figure 2). 

In the present study we applied a self-designed interview method. Besides analysis of publicly available papers, press releases, newspaper articles and the blogosphere, face-to-face expert estimations were made with 24 stakeholders related to the field of Hungarian catering. This series of preliminary interviews were conducted with the purpose of determining the set of relevant actors and interests. The interviews were carried out in 2015 and 2016. The aim of this preliminary phase of interviews was to outline the most important stakeholder groups and the set of the potential objectives of the stakeholders identified in the Appendix A. As a result of these preliminary investigations, a robust and relatively well-manageable set of actors and goals could be identified. In setting up a pool of experts a specific procedure was followed. In this phase we pursued the following logic. We considered experts to be people (1) who have a direct “field” experiences in catering functions as parents or teachers; (2) people whose job directly involves a catering business with relatively long experience in the practice of SC and whose existence directly depends on this enterprise S; (3) independent experts, preferably those who have been especially active in professional social debates concerning the catering regulations in the printed and electronic media; (4) experts who have been actively involved in the preparation and enforcement of the new regulatory framework of the SCS. The attitude of experts towards school catering has not been taken into consideration, neither in the choice of experts, nor in the interview phase.

The second phase of research was a semi-quantitative interview. The list of potential participants was collected on the basis of intensive research into publications (including professional conferences, various formal and informal meetings of professional communities, the blogosphere and the grey literature), membership of professional organizations and the personal recommendations of other experts. 

In summary, the names of 321 experts were collected (not including parents and teachers). Out of these experts, we tried to make contact with specialists who supposedly, in the opinion of at least two members of the community of authors, have a more ‘holistic’ approach to the SCS universe without taking into consideration their attitude to the SCS question. In the process of our study, all relevant stakeholders were taken into consideration. In this way, 78 experts were selected. We contacted 61 of them; 45 respondents expressed their willingness to participate in the research. Due to time and financial constraints, 33 expert-interviews were carried out, all of them face-to-face. Additionally, we interviewed 26 parents and 13 teachers. Specific attention was paid to choose parents and teachers from relatively well-off and less developed regions of Hungary: cities like Budapest and Szeged, a small town (Hajós) and a village (Báta). Personal acquaintance played an important role in the choice of teachers and parents. The characteristic features of interviewees are summarised in the Appendix A.

The quantification of the intensity of actor–actor influences, as well as the actor–objective relations has been developed in a step-by-step manner. As we have experienced with our previous research [47], filling out the input matrices in the form of MS Excel worksheets for research was a very time-consuming (and in some cases a rather boring) process often leading to internal contradictions because it was very difficult to achieve a general common interpretation of different scales. That is why a semi-structured interview was used [48]. The conversion of the verbal estimations was carried out in the personal interview phase with the help of the researchers. The only task of the researchers was to help interpret the different scales. This method proved to be an efficient method for achieving internal consistency in the input data for analysis [49].

In the framework of the interviews, we asked the respondents to evaluate the bargaining power of each actor in comparison with another actor (e.g., Government vs. Teachers, Government vs. Catering service managers etc…) on a 0–4 scale. The interpretation of this scale was the following: 0—no direct influence.1—actor A can eliminate the tactical steps of actor B.2—actor A can jeopardize/eliminate the projects of actor B.3—actor A can jeopardize/eliminate the strategic goals of actor B. 4—actor A can substantially influence/dominate actor B. 

In the second phase of interviews, we asked the interviewees to evaluate the attitudes of actors (stakeholders) towards different elements of goal set on a −4… 0 … +4 scale. The interpretation of the scale was as follows:−4 the objective is against the vital interest/jeopardizes the existence of the actor −3 the objective jeopardizes the strategic mission of actors −2 the objective jeopardizes the tactical goals of the actors−1 the objective jeopardizes the operative goals of the actor 0 the actors’ attitude towards the goal is neutral +1 the objective falls in line with the operative goals of the actor +2 the objective falls in line with the tactical goals of the actors +3 the objective considerably supports the strategic goals of the actor+4 the objective is a vital interest of the actor

The participants received the cumulated input-matrices and their interpretation by e-mail, and had the opportunity to suggest some modifications. The results of the MACTOR analysis were discussed in detail with a representative pool of respondents in a group discussion and in face-to-face interviews. This phase of the research was an explorative one since our ambition was not to create a representative sample but rather to collect a relatively wide range of opinions.

The research topics were considered very important and interesting questions by all participants, which is why the willingness to participate in the interviews was very high. People evaluated their participation in the research very positively and they were cooperative. Participants were willing to share their experiences and views with others on different problems related to the school catering system development.

### 2.3. Ethics

The Inter-Faculty Research Ethics Committee of the faculty of Budapest Corvinus University approved both the concept and procedure of the research (Ref. No.15/12/2014). All of the participants signed an informed consent, which described the procedure of the research in detail.

## 3. Results

### 3.1. Identification of Key Actors

The framework of individual comprehensive interviews with 24 experts offered a favourable opportunity for understanding the views of different expert actors and their key goals. It was very interesting to see the convergence in the opinions of the respondents concerning the estimation of the goal structures of different actors.

#### 3.1.1. Governments

All respondents agreed that the role of government was essential in the SCS, and that neither the national nor local governmental organs consider the development of the SCS as a priority because it is much less spectacular (i.e., it offers much less possibility to increase the number of votes) than the delivery of a new sports complex or Christmas gift for older voters. This rather curious behaviour is an integrated part of the paternalistic political culture of Hungary [50]. This phenomenon can serve as an explanation of the fact that up to 2010 no significant efforts were made to change the rather negative trends of obesity among children and adolescents. 

At the same time, the efforts of the government to change the traditional SCS in Hungary were weakened by the differences of approaches of various ministries and public administration institutions. The attention of governmental organs was divided among different goals and dispersed projects aiming at changing the SCS in Hungary. A possible explanation for this is the scattered structure of the Hungarian public administration and governmental system: the SCS, as a catering service, is supervised by the state secretariat of Health Care at the Ministry of Human Capacities; as a service operated in schools, it is subordinated to the state secretariat of Public Education of the same ministry; as a part of the food chain, it is controlled by the Ministry of Agriculture. Under these conditions a wide range of actions were taken by different actors and lobbies. In the opinion of our interview partners, the SCS is regarded as an important market for local products by the Hungarian Ministry of Agriculture. It is not a coincidence that that the ministry intensively supported the Canteen-Pattern (in Hungarian: Minta Menza^®^) project, which aims to sell local products in the SCS, often involving specific ingredients (e.g., game-meat, locally grown mushrooms, quail eggs). The project mainly focused on increasing the diversity of dishes and marketing [51]. Children were considered in this framework as consumers, and not as real partners in development. Another state-supported program was the Canteen Reform (in Hungarian: Mensareform^®^) project developed by some Hungarian celebrity chefs. The goal of this project was the diversification of dish portfolios in the SCS, streamlining the traditional dishes by the introduction of such dish names as “bang-bang crazy chicken” as well upgrading the knowledge of local chefs by a sophisticated qualification system. Both of these programs lost their impetus in the absence of government support [52].

The question arises of what the reason was for launching different programs in parallel. In the opinion of our interview partners, the reason is not just the activity of different lobbies and governmental organs: each public servant tries to highlight his/her importance with the management of one or more programs. For that reason, numerous small-scale projects were initiated, often without any real chance of accomplishment (e.g., “Start with breakfast!”, or “Happy week” to increase tap water consumption).

Under these conditions the initiative of the Healthcare secretariat was not able to mobilize the agricultural lobby, nor the catering system providers.

#### 3.1.2. Local Authorities

Local authorities have played an important role in the development of the SCS because the operative running of schools is their responsibility. All of the respondents agreed that (1) the decisions, taken by local authorities concerning the SCS are considerably influenced by policy, namely by the central government and local lobbies, (2) local authorities have a relatively high level of influence on business enterprises, because they can allocate additional financial resources to increase the per-capita financial subsidy for school catering by agreements with SCS firms, (3) the SCM plays an important role in the welfare system as under these conditions the relative availability of food and school services can be considered a political question par excellence.

#### 3.1.3. The Catering Service Providers 

There is an increasing number of municipalities that buy the catering service for schools from specific enterprises, which run the kitchen at schools or run the finishing kitchens, where they heat and serve the meals prepared in the central kitchens of the enterprises. The catering service provision has been considered a flourishing business.

#### 3.1.4. Catering Service Managers

The catering service managers are the bosses of kitchens or finishing kitchens. In general, their work consists of the procuration of raw materials, food preparation and management of serving process.

All respondents agreed that local managers are extremely important in the SCS but their scope of decision is rather limited due to budget limits and the difficult systems of regulations. A surprising phenomenon has occurred, namely the burnout the catering managers.

A catering manager said: “I am sick and tired of hearing from early morning to late night each day the snivelling spoiled children. They encourage each other to refuse to eat our food, and they complain about the bad meals we prepare.”

The catering managers are frontline solders of the system; however, they felt abandoned. They complained in the following way: “From these limited material resources (i.e., a lack of money) we are not able to buy the raw materials for the kind of food which could satisfy the requirements of Decree… I do not know enough recipes to prepare a diverse menu to satisfy the requirements—we did not receive any help to do this.”

#### 3.1.5. The Parents

Our interviews highlighted that in numerous cases parents do not have enough time to cook, and often even go to fast-food restaurants. This phenomenon decreases families’ influence on the healthy nutrition of children and adolescents. All of the respondents agreed that due to the considerable differences within the socio-economic structure in Hungary, it is hard to speak about parents as a homogenous group, because (1) there are parents who simply do not have energy/time to care about the food consumption of their children. There are those (2) who worry about the low quality/quantity of food served to their children in school canteens, which is why they pack them sandwiches or give them money to buy additional food, while (3) in the case of poor families the school canteen plays an important role in relieving families (and their budgets) from the burden of daily food provision. It was a very frequent reflection that: ”There is so much talking about school feeding but this is the first time that my opinion is important for someone”. Or, as one parent formulated: “This is a very long-lasting problem and a crucial issue but lack of money and attention is an obstacle to improve the situation, so there is not too much to expect”. One school canteen manager said: “The most important days for us are Mondays and Fridays: on Monday we have to prepare to energy-rich food, because the children wants nutrient rich food after the unsatisfactory food consumption in the weekend, and on Friday we have to “fill up” the children with copious food”.

#### 3.1.6. The Children

Our interviews highlighted that the current world of the SCS is quite distant from the demands of children. There is old, adolescent-sized furniture and sometimes rude and un-motivated kitchen staff (mainly older, often burnt-out female employees), who acquired their experiences in years of relative food shortage and are not able to communicate in an appropriate way with children. Catering reform will only be successful if pupils like and choose to eat these meals, but this aspect has been neglected in Hungary.

The majority of specialists agreed that time has passed the old-style catering infrastructure by, and it has not been renovated for decades. This contrasts noticeably with the vivid colours and modern interiors of the majority of fast-food restaurants which target the younger generation. Under these conditions there is just a relatively low chance of attracting young consumers, who often consider traditional food as old-style.

Children were not included in our evaluation/interview because we focussed on the socio-economic arena of SCS. The primary aim of our investigation was the determination of optimal (qualitative and quantitative) parameters of school meals.

#### 3.1.7. Teachers

Our interview partners agreed that, theoretically, teachers play a very important role in the formation of the eating behaviours and eating habits of children. The teachers eat mainly the same food as the children, if they are eating in the same canteen.

Under these conditions, there is an increasing tendency towards overburdened, burnt-out teachers. As one teachers formulated it: ”I have a lot of problems in school and in my private life, I am simply too tired to deal with such problems as the nutrition of the children”. As another formulated it: “I am fed up with the fact that society tries to push all its problems onto the schools and teachers. I do not feel it to be my responsibility to care about what children eat under such conditions, when I have to teach them the most elementary rules of social behaviour, just because their parents are playing with their smartphones or lingering on social websites.”

### 3.2. Analysis of Actors’ Positions and Strategies Using the MACTOR Method

By using the results of extensive interviews, we determined the set of key actors, and the set of strategic goals, which were determined for one or more actors.

The matrix of the direct influences of actors is shown in Table 4. Based on the actor–goal matrix, we have depicted the position of different actors by the analysis of correspondence [53]. This method is widely applied for the visualization of the relative position of different actors to each other. 

The goals in this table are based on preliminary interviews. This is a rectangular matrix. The main diagonal of the matrix (the influence of a given actor on itself) is by definition zero. The actor considered as the influencer is placed in the row, and the influenced party is situated in the column. The values in the cells of matrix are the simple averages of responses obtained as a result of discussion. To be on the safe side we have calculated the averages by group of actors (e.g., teachers, parents, catering managers etc.), but we have not been able to prove significant differences between the two methods. The in-depth analysis of standard deviations according to interviewees could go beyond the limits of the current research.

Some remarks about the results of the matrix of direct influences based on in-depth interviews are indicated as follows:

—The matrix highlights the considerable influence of the government on the behaviour of catering service providers and on the SCS in general, because the government has a considerable influence on the monetary resources for school catering.

—Local authorities can exercise an important influence on catering service providers because they have the possibility to select the SCS provider for different schools in the framework of the public procurement procedure. 

—In the opinion of interview partners, a basically positive tendency has occurred over the last years: the increasing influence of parents’ organizations on the life of schools. At the same time, due to the lack of democratic roots and traditions it is quite difficult to achieve a situation in which parents’ influence is based on an absolute majority of the parents and not just on a relative majority of a small group with active and noisy members. Due to the lack of democratic roots and traditions, it is quite difficult to achieve parents’ influence based on an absolute majority of the parents and not just on a small group with active and noisy members. The interest relation of the different actors is summarized in Table 5. This is a not a rectangular matrix. The actors are placed in rows and the different goals are located in columns. The values in the cells of matrix are the simple averages of responses obtained as a result of discussion. 

The health of children is a generally accepted goal for all participants, but this goal is relatively more distant in time, which means it is difficult to translate into operative actions. Local authorities are in direct connection with the population, so the taste of food is especially important for them. The children’s acceptance of food is important for the parents, too, because in this way they can reduce their household expenditure on feeding their children. A well-fed child can be managed easily so the taste and acceptance of food is important for teachers, as well. It should be highlighted that cost minimization is quite an important question for the majority of actors.

According to our findings, school feeding was important for parents. It can be assessed as a positive point, that, at least on the verbal level, the healthiness of school feeding was evaluated as a question of great importance. This can be considered as a favourable tendency. The bargaining position of different actors has been depicted on a two-dimensional coordinate system. On one ordinate of the system, we have indicated the level of influence of the given actor. This value has been calculated based on Table 4 by the summation and normalization of direct influences for each given actor in relation to another actor [47]. The dependence of actors has been calculated in a similar manner. 

Analysing the map of influences and dependences between actors (Figure 3), it is obvious that the government has a relatively favourable bargaining position because it has a relatively high level of influence and a low level of dependence. The direct socio-economic environment of children’s food consumption is the following: the triangle of teachers, local authorities and parents have roughly the same position, namely a relatively high level of influence and a low level of dependence. The owners of the SCS firms have approximately the same level of dependence as the former three actors, with a much lower influence. The two key actors of the SCS system, the children and the catering service managers, have an extremely low level of influence, which—especially in the case of the children—is accompanied by high dependence. In other words, the two critical actors of the systems, namely the actual service providers and the children, have the least possibility to influence the operation of the system.

The analysis of influence dependence matrix offers information on estimated power relationships between actors and also on the way of thinking of respondents. It can be considered as a rather negative tencency that interviewees evaluated the catering system as a set of different, relavitevly separated actors instead of a coherent system with numerous relations. This fact is experessed in a high number of zeroes indicating no infuence.

The indicator “mobilizing force” of different goals has been calculated based on the acceptance of differing goals weighted by bargaining power (influence) on actors.

Analysing the mobilizing force of the different interests, good taste and children’s health have the highest value (Table 6). Notwithstanding, it is important to stress that the mobilizing force of the cost minimization of health expenditure promotion and simplicity is much higher than the healthiness of food criteria.

## 4. Discussion and Conclusions 

The results of this survey highlight that government support has been rather weak so far because the majority of parents accept the importance of healthy eating, but they avoid the conflicts with their children. The overburdened teachers have done little to change the eating habits of the students. The caterers and the catering service managers support the best solution for themselves. At the same time the direct influence of catering managers is relatively low; however, their behaviour and attitude are of great interest in any reform. It is a contradiction that they have not been prepared to make use of these reforms. Neither the government nor the local municipalities have been determined enough to mobilize additional financial and intellectual sources to change the situation and consequently follow the policy. In general, it can be stated that the MACTOR method has been an efficient tool to uncover the direct and indirect force relations and motivations of different actors. 

Our results are consistent with conclusions drown by another actors. For example, in the opinion of Gaál et al. [54], the central government of Hungary has almost exclusive power to formulate and realize strategic decisions and shape the regulatory framework of the health care system, as well as make public-health-related interventions. The majority of schools (with the exception of schools owned by foundations and churches) are in state ownership.

Due to a wide agreement on the long-term instability of the Hungarian health-care system [55,56,57,58], preventive measures should be given priority.

It is a contradiction that while all Hungarian governmental programs [59,60,61,62] have highlighted the importance of prevention in the health-care system, obesity among the young population groups has been increasing in the last few decades [63,64].

It should be highlighted that neither schools, teachers, nor parents can be considered homogenous groups [65]. There are significant differences in parents’ attitudes and behaviour in relation to the school catering system. The results of van Zenten [66] and Raveaud and van Zanten [67] highlight that upper- or middle-class parents have a much higher level of aspiration to participate in the decision-making process in the framework of the school than working-class parents, who are more ‘loyal’ to local schools [68].

The importance of school feeding for parents was a surprising result. It can be regarded as positive that the healthiness of school feeding has been evaluated as a question of great interest. This can be considered a favourable tendency.

The results of investigations have demonstrated clearly that children are key players in the Hungarian SCS but have little influence on the system. In Hungary, there does not exist any mechanism to survey the opinion and test the preference system of children. The most reliable indicators are the cleaners, who can furnish some information on the quantity of food and the amount leftover. There are neither experiences nor resources or mechanisms to uncover the drivers behind children’s behaviour [69].

The average uptake level is low, varying from 40% to 95%; the highest take up rate in Europe is in Finland (over 90%), probably because school lunch plays a central role in education [70].

Based on our analyses, the failure of the new regulation of the SCS was predictable because children have not accepted dishes with low salt content and milk with low fat content. The uptake level of the reform was extremely low as children were not involved in the new and rapidly introduced changes. The reluctance of children induced a chain reaction: neither the parents nor the teachers were motivated, nor did they have any background knowledge to argue for the positive aspects of the reform. The catering service providers and the local managers experienced a high level of food waste. These negative views were reinforced by the media, which over-emphasized the negative aspects of the new regulations. Under these conditions, it was relatively easy to forge a coalition between parents, catering service providers and managers, teachers and local authorities to choose the easier option: i.e., to force the government to substantially modify the original legislation. The fate of this regulation, based on sound professional arguments, lends itself to an analysis of the failures leading to the collapse of the regulation. The most important failures can be summarized as follows:

1. There was no general, well-designed study and testing of the effects of the regulation in the framework of a pilot study.

2. There was no multilateral communication between the government and the stakeholders, mainly with catering service providers and catering managers.

3. The teachers did not have the necessary background to discuss the issue with children due to their total illiteracy in the field of nutrition.

4. Children do not have to learn practically any nutrition or food skills during their studies in the primary and secondary school, so they did not have the capacity to understand the reason for the changes.

5. The energy and resources of different governmental organs were scattered among different, partially competing, goals.

The school lunch provides an excellent opportunity to learn healthy eating habits, promote nutrient-rich foods, involve children in foodservice planning and improve the nutritional intake of school children. There is an urgent need to improve the situation, taking into account the WHO tool for the development of school nutrition programmes in the European Region [71]. The policy paper emphasizes that healthy food and nutrition should be given a high priority on every school agenda and school meals are an indispensable part of a whole school approach to health promotion. To ensure effective implementation, stakeholders should review the available information on nutritional status and eating patterns before developing school lunch standards and issuing any regulation on nutrition requirements in public food-services.

As a conclusion of this study, it can be determined that the position and the system of interests of different actors must be taken into consideration before setting up SCS-related regulations. Children play a central role because they will be able indirectly influence parents and teachers. From this it follows that considerable resources should be mobilized to understand the driving forces of children’s behaviour and their taste. On the other hand, from a retrospective perspective it is obvious that the government wanted to improve the SCS “by force”, following top-down logic, with minimal mobilization of material and human resources on education and widening the possibilities of raw material procuration.

Cooperation among all the different stakeholders is crucial when working out a school food and nutrition policy [72]. Moreover, one can even state that cooperation between different players, including coopetition, is a fundamental issue in today’s world [73,74]. 

Representatives of teachers, parents, pupils, caterers and representatives from a school’s governing body should develop an action plan and introduce a comprehensive education program. The study of policy itself can be an educational tool. Teaching about food and nutrition policies in schools would make students active participants throughout the policy process [75]. The SCS should be better integrated into the general education program. For example, the participation of children in food preparation and servicing could promote family-life education and a more harmonious division of work between the genders, too. In order to establish a successful and effective strategy, a continuous dialogue is needed between parents, health promotion experts, teachers and their organizations. It should be emphasized that there is no universal solution: the specific needs of children should be taken into consideration because a good school meal is an investment in the future.

## Figures and Tables

**Figure 1 nutrients-11-00716-f001:**
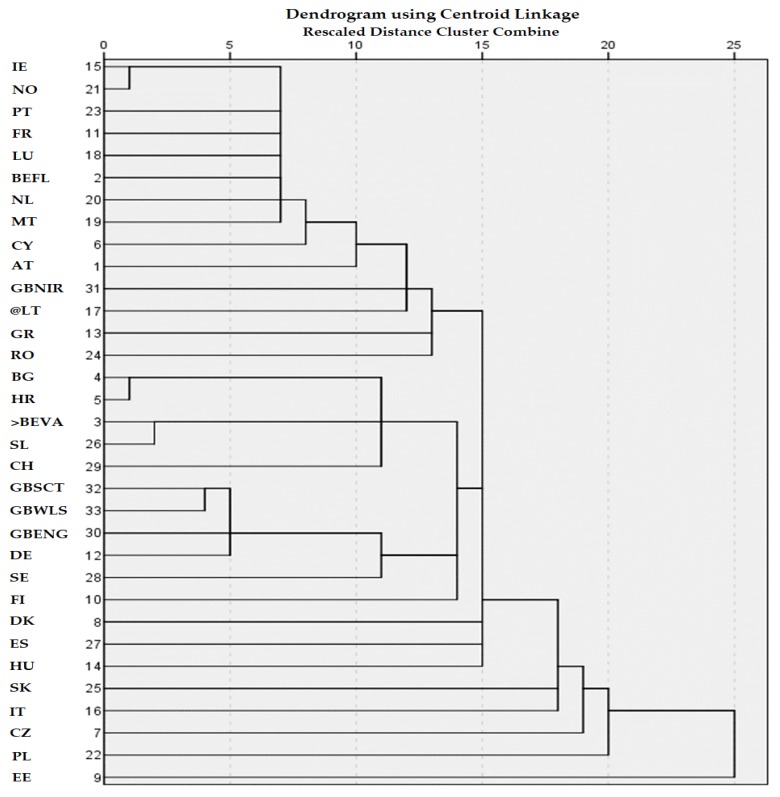
Results of cluster analysis of different EU member states based on their school feeding policy. Source: Authors’ own construction, based on EU [28].

**Figure 2 nutrients-11-00716-f002:**
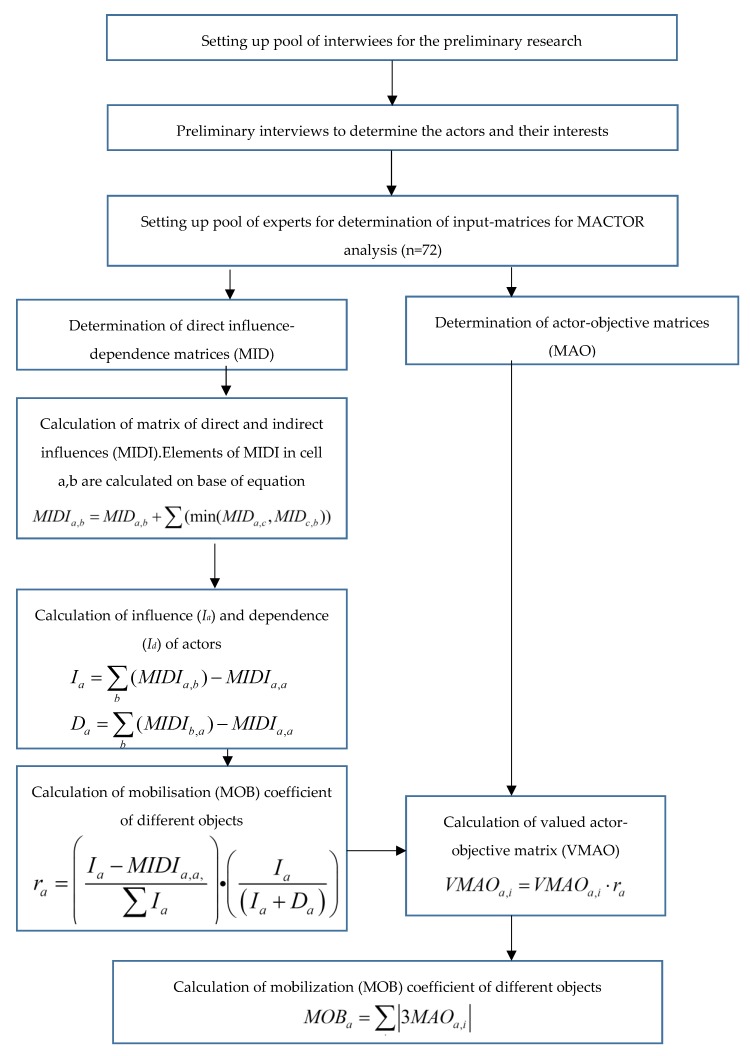
Flowchart of investigations.

**Figure 3 nutrients-11-00716-f003:**
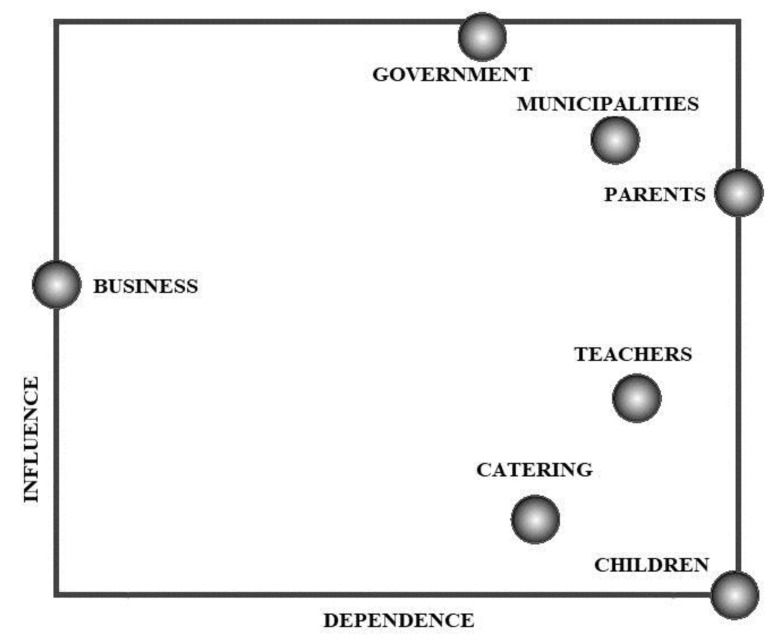
The influence–dependence relations of actors.

**Table 1 nutrients-11-00716-t001:** Summary of the surveys of school catering systems.

Characteristic Features/Indicators	Canteen Panorama 2008	Canteen Panorama 2013	Canteen Panorama 2017
**No. of schools (educational institutions)**	3099	260 representatively chosen	139 elementary schools
**Elementary schools in the sample (%)**	62	62	100
**Secondary schools in the sample (%)**	17	31	
**Elementary and secondary schools (%)**		7	
**Schools offering warm meal at least once a day with canteen (%) ***	92	17	100
**School milk program (%)**	15	35	72
**Free fruits and vegetables (%)**	14	78	95
**Free drinking water outside the bathroom (%)**	36	58	75
**Proportion of students eating at the school canteen (in %)** **Primary (7–10 years)**	85	87	88
**Lower secondary (%) (10–14 years)**	47	63	61
**Secondary school (%) (14–18 years)**	20	27	
**Survey of children’s satisfaction conducted by the schools (%)**	29		
**Qualified food catering manager (%)**	76.5	83	no data
**Food planning with involvement of a dietitian (%)**	10	29	38
**Energy and nutrients content calculation (%) ****	22	57	no data
**Net food budget calculated for raw material *****	0.72 EURO	1.2 EURO	1.15 EURO
**School snack bar in institutions (%)**	45	50	44
**Vending machines (%)**	30	34	no data

* In each elementary school, and proportion of secondary schools without boarding; ** Energy and nutrient content calculation is based on official nutrient and energy content information, the age-specific menu is calculated based on this information and in some cases even a specific menu planning sotftware is applied. *** Hungarian currency converted to EURO on yearly average conversion rate. Source: Authors’ own construction, based on Martos [30], Bakacs et al. [32] and Bakacs et al. [31].

**Table 2 nutrients-11-00716-t002:** The energy limits of school feeding *.

	1–3 years	4–6 years	7–10 years	11–14 years	15–18 years
**Boarding school**	4600–5450	5643–6900	7100–8600	8360–10000	8360–10,900
**Nursery**	3340–4000				
**Three meals/day**		3800–4600	4600–5500	5500–6500	5400–7100
**One meal/day**		1880–2500	2500–3200	3000–3500	3000–3800

* Original values in kcal, converted to KJ and rounded. Source: EMMI [38].

**Table 3 nutrients-11-00716-t003:** The most important changes in the public catering decrees.

	2014 *	2016 **
	Food-Based Standards	Food-Based Standards
**Specific foods and food groups have to be provided daily for all age groups (for one person)**		
**5 meals/day** **(in a boarding institution the public caterer is obliged to offer main meals three times and two snacks two times)**	4 portions of fruits or vegetables per day, at least one of which should be raw ***	4 portions of fruits or vegetables per day, at least one of which should be raw
	3 portions of cereals, at least one which should be whole grain	3 portions of cereals, at least one which should be whole grain
	0.5 l milk or a diary product with an adequate amount of calcium	Removed
**Nursery (1 to <3 years)** **(75% of the daily energy requirement should be covered by two main meals and two snacks)**	3 portions of fruits or vegetables per day, at least one of which should be raw	3 portions of fruits or vegetables per day, at least one of which should be raw
	2 portions of cereals, at least one of which should be whole grain	2 portions of cereals, at least one of which should be whole grain
	0.4 l milk or a diary product with an adequate amount of calcium	Removed
**3 meals/day** **(by offering boarding 65% of daily energy requirement should be covered by one main meal and two snacks)**	2 portions of fruits or vegetables per day, at least one of which should be raw	2 portions of fruits or vegetables per day, at least one of which should be raw
	2 portions of cereals, at least one of which should be whole grain	2 portions of cereals, at least one of which should be whole grain
	0.3 l milk or a diary product with an adequate amount of calcium	Removed
**1 meal/day** **(offering one main meal (dinner), 35% of daily energy content)**	1 portion of fruits or vegetables per day, at least three of which should be raw over a 10-day catering period	1 portion of fruits or vegetables per day, at least three of which should be raw over a 10-day catering period
**Supplementation**		If pre-primary, 5 or 3 meals are provided a day, milk or a diary product with an adequate amount of calcium should be served every day
**Regulations, limitations and prohibitions of using certain foods and products**		
**Fat content of milk**	2.8% or 3.6% milk fat milks should be served for age group 1–3 years; 1.5% or <1.5% milk fat milks should be served above 3 years old	2.8% or 3.6% milk fat milks should be served for age group 1–3 years; 2.8% or <2.8% milk fat milks should be served above 3 years old
**Water**	Constant access to fresh water (outside of bathrooms)	Constant access to fresh water (outside of bathrooms)
**Added sugar**	Free sugar should not exceed 8% of total energy in a 10-day catering period	Free sugar should not exceed 10% of total energy in a 10-day catering period
**Salt and free sugar**	Salt and sugar should not be placed on dining table	Salt or sugar containers should be labelled: “Excessive salt intake could cause cardiovascular diseases, obesity and diabetes!”
**Salt content**	Daily salt intake should be reduced to 5 g/day up to 1st of September 2021 At the age bracket 7–10 years salt intake should be reduced to 3.5 g/day up to 1st of September 2021. If the institute offers one meal/day the salt content of the main meal should be reduced up to 2 g/day at age bracket 7–10 years up to 1st of September 2021	

* Public Catering Decree EMMI (Ministry of Human Capacities), Decree 37/2014 (came into force in September 2015). ** Decree on modification of EMMI (Ministry of Human Capacities), Decree 37/2014 (came into force in December 2016). *** The ration of the raw fruit/vegetable requirement is not included in the legislation but it is assumed that this can be explained by higher vitamin content of these goods [39]. Source: MHC [34] and MHC [37].

**Table 4 nutrients-11-00716-t004:** The matrix of direct influences on actors measured on a 0–4 scale (0—no direct influence, 4—very strong influence).

	GOV	MUNICIP	PARENTS	CHILDREN	MANAGERS	BUSINESS	TEACHERS
Government GOV	0	3	1	0	1	4	2
Local authorities MUNICIP	1	0	1	0	1	4	0
PARENTS	1	3	0	3	0	0	2
CHILDREN	0	0	2	0	0	0	1
Catering service managers MANAGERS	0	0	0	2	0	0	1
Catering service providers BUSINESS	1	1	0	0	3	0	0
TEACHERS	1	2	2	3	1	1	0

**Table 5 nutrients-11-00716-t005:** The actors’ interest relations measured on a −4…+4 scale.

	Good Taste of the Meal (TASTE)	Healthiness of the Food (HEALTH)	Healthy Children (HEALTHYCHI)	Vote maximisation (VOTE)	Feeling of Being sated (SATED)	Minimisation of Expenditure on Health Promotion (CPOSTMIN)	Simplicity of Food Preparation (SIMPLE)
**GOV**	1	3	4	4	3	3	0
**MUNIICIP**	2	1	4	4	3	4	0
**PARENT**	4	3	4	0	2	3	0
**CHILD**	4	1	4	0	2	0	0
**CATERING**	3	0	0	0	2	4	4
**BUSINESS**	3	0	0	0	2	2	4
**TEACHERS**	3	1	3	0	3	1	0

Interpretation: −4 the objective is against the vital interest/jeopardizes the existence of the actor, +4 the objective is a vital interest of the actor.

**Table 6 nutrients-11-00716-t006:** Mobilising force of different goals.

Goal	Mobilising Force
**good taste of the meal**	12.8
**healthy food**	17.1
**healthy children**	17.8
**vote maximisation**	8.6
**feeling of being sated**	10.2
**minimisation of expenditure on health promotion**	17.4
**simplicity of food preparation**	2.4

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
