# Peer review of "The Reform of School Catering in Hungary: Anatomy of a Health-Education Attempt"

_nutrients, 2019, doi:10.3390/nu11040716_

Round 1

Reviewer 1 Report

Review of “The Reform of School Catering in Hungary: Anatomy of a Health Education Failure” by Anna Kiss, József Popp, Judit Oláh, Zoltán Lakner

I find the general idea of the paper, that there needs to be more coordination among the various actors in order to reform school lunches in Hungary, interesting and important. And the approach of interviewing many of the key actors involved in reforming school lunches is appropriate, though the 33 interviews seems a bit low to learn very much (line 157). I am a bit confused why there were only 18 observations (line 175) but Table 12 contains information on 24. And also the missing data (e.g. not all groups have the same number of observations -- e.g., parents in table 2 have only 22 places of residence) . 

I do recognize that the authors think of this as an exploratory study,  but I would like a bit more detail about the data. For example, are there measures of interviewee burden?  Cooperation?  Failure to complete?

Did your data analysis reveal anything surprising?  Since this was an exploratory study, what did you learn that you had not anticipated?

I believe the thought about how children are key players here but have little input could be crucial.  Can you develop that point a lot more? How can this type of analysis help do make that point?

I am a bit confused about Figure 1 (discussed lines 371-380). What precisely is “INFLUENCE”? DEPENDENCE?  I suspect influence is how well they can impact the food offerings in the schools, but dependence is not that straightforward.  Is dependence just a measure of how each actor/group relies upon the others in the scenario?  On line 376, you state that the influence of the BUSINES is much lower than the others, in the previously discussed 3-group, but clearly they have higher influence than teachers on the graph.           

In the discussion of Table 6, you state (line 386) that good taste and children’s health have “the highest values,” but healthy food has a higher score than good taste, as does minimisation of expenditure on health promotion.  So this statement seems incorrect.  And then the analysis takes ”dimensionless” numbers and adds them together (lines 387-8) and compares them to other sums. Do they really work that way in some understandable sense?  

I do not understand the axes of the 2 dimensions in Figure 2 (lines 393-395). Are they some kind of factor loadings or principal components?

In the Discussion, you highlight 5 key points (lines 413-423), yet your analysis only seemed to deal directly with the last one (line 422-3).   

This last issue brings up the key point about what is the precise contribution of this paper, and it was not clear to me what the paper did to change the landscape of how to think about making changes to school lunches to make them more healthful.  But I suspect that your analysis did do something, perhaps a lot, in that regard. It would be useful to make fairly specific statements about the contribution of this research.  Let us know explicitly (though briefly), the state of knowledge before your analysis, what your analysis adds to this, and what your analysis suggests are important next steps that should be taken to understand better the process of making successful changes. You highlight this as an exploratory study, so let us know explicitly what you found and how this will help make progress.

Overall, I am a bit confused about the goal of this paper.  Is it to look at some “exploratory” data and then make policy recommendations? Or is it to provide a better foundation for understanding the processes that need to happen for successful implementation of school lunch reforms. I suspect you are mixing the two together, but they really are separate goals and should be presented as such.

Please define the term SCM.

Author Response

Reviewer 1

The reviewer’s useful comments and suggestions are highly appreciated. We have revised the manuscript in accordance with the reviewer’s comments. We have revised the manuscript in accordance with the reviewers’ comments. We have revised the Abstract, Introduction, Materials and Methods, Results, Discussion and conclusions sections, and References, rewritten many parts, and searched for and updated literature and provided additional information on the reform of school catering in Hungary.

Overall, we have made major revision to this manuscript. Firstly, the references have been updated and expanded. We carefully selected 62 papers, which are considered important or innovative studies, or comprehensive reviews offering us a wider picture of the reform of school catering in Hungary. In the new version of the manuscript, 13 references have been added. Secondly, the structure and sections have been revised, subsections have been added and several new paragraphs, tables and figures have been inserted. This version now includes revised Abstract, Introduction, Materials and Methods, Results, Discussion and conclusions sections, and References. Revised sections and subsections have been carefully reorganized, and new and modified paragraphs added to each section. Thirdly, language and grammar have been edited under the guidance of a professional native speaker.

Thank you very much for your comments and for taking the time to thoroughly review the manuscript. Your comments have been very constructive in helping us to clarify our message.

Comments and Suggestions for Authors

Review of “The Reform of School Catering in Hungary: Anatomy of a Health Education Failure” by Anna Kiss, József Popp, Judit Oláh, Zoltán Lakner

I find the general idea of the paper, that there needs to be more coordination among the various actors in order to reform school lunches in Hungary, interesting and important. And the approach of interviewing many of the key actors involved in reforming school lunches is appropriate, though the 33 interviews seems a bit low to learn very much (line 157). I am a bit confused why there were only 18 observations (line 175) but Table 12 contains information on 24. And also the missing data (e.g. not all groups have the same number of observations -- e.g., parents in table 2 have only 22 places of residence).

Response

Thank you for your useful comments. We have corrected data and presented the final numbers in the modified version of our manuscript (see track changes).

The data collection for the analysis was a multiphase process. In the present study we applied a self-designed interview method. Besides analysis of publicly available papers, press releases, newspaper articles and the blogosphere, face-to-face expert estimations were made with 24 stakeholders related to the field of Hungarian catering. This series of preliminary interviews were conducted with the purpose of determining the set of relevant actors and interests. The interviews were carried out in 2015 and 2016. The aim of this preliminary phase of interviews was to outline the most important stakeholder groups and the set of the potential objectives of the stakeholders. As a result of these preliminary investigations a robust and relatively well-manageable set of actors and goals could be identified. In setting up a pool of experts a specific procedure was followed. In this phase we pursued the following logic. We considered experts to be people (1) who have a direct “field” experiences in catering functions as parents or teachers; (2) people whose job directly involves a catering business with relatively long experience in the practice of SC and whose existence directly depends on this enterprise S; (3) independent experts, preferably those who have been especially active in professional social debates concerning the catering regulations in the printed and electronic media; (4) experts who have been actively involved in the preparation and enforcement of the new regulatory framework of the SCS. The attitude of experts towards school catering has not been taken into consideration, neither in the choice of experts, nor in the interview phase.

The second phase of research was a semi-quantitative interview. The list of potential participants was collected on the basis of intensive research into publications (including professional conferences, various formal and informal meetings of professional communities, the blogosphere and the grey literature), membership of professional organizations and the personal recommendations of other experts.

In summary, the names of 321 experts were collected (not including parents and teachers). Out of these experts we tried to make contact with specialists who supposedly – in the opinion of at least two members of the community of authors – have a more ‘holistic’ approach to the SCS universe without taking into consideration their attitude to the SCS question. In this way 78 experts were selected. We contacted 61 of them; 45 respondents expressed their willingness to participate in the research. Due to time and financial constraints 33 expert-interviews were carried out, all of them face-to-face. Additionally, we interviewed 26 parents and 13 teachers. The characteristic features of interviewees are summarised in Supplement 1.

The quantification of the intensity of actor-actor influences, as well as the actor-objective relations has been developed in a step-by-step manner. As we have experienced with our previous research [37], filling out the input matrices in the form of MS Excel worksheets for research was a very time-consuming (and in some cases a rather boring) process often leading to internal contradictions because it was very difficult to achieve a general common interpretation of different scales. That is why a semi-structured interview was used [38]. The conversion of the verbal estimations was carried out in the personal interview phase with the help of the researchers. The only task of the researchers was to help to interpret the different scales. This method provided to be an efficient method for achieving internal consistency in the input data for analysis [39].

In the framework of the interviews we asked the respondents to evaluate the bargaining power of each actor in comparison with another actor (e.g. Government vs. teachers, Government vs. Catering service managers etc…) on a 0-4 scale. The interpretation of this scale was the following:

0 – no direct influence

1 – actor A can eliminate the tactical steps of actor B

2 – actor A can jeopardize/eliminate the projects of actor B

3 – actor A can jeopardize/eliminate the strategic goals of actor B

4 – actor A can substantially influence/dominate actor B

In the second phase of interviews we asked the interviewees to evaluate the attitudes of actors (stakeholders) towards different elements of goal set on a -4… 0 … +4 scale. The interpretation of the scale was as follows:

-4 the objective is against the vital interest/jeopardizes the existence of the actor

-3 the objective jeopardizes the strategic mission of actors

-2 the objective jeopardizes the tactical goals of the actors

-1 the objective jeopardizes the operative goals of the actor

0 the actors’ attitude towards the goal is neutral

+1 the objective falls in line with the operative goals of the actor

-2 the objective falls in line with the tactical goals of the actors

3 the objective considerably supports the strategic goals of the actor

4 the objective is a vital interest of the actor

The participants received the cumulated input–matrices and their interpretation by e-mail, and had the opportunity to suggest some modifications. The results of the MACTOR analysis were discussed in detail with a representative pool of respondents in a group discussion and in face-to-face interviews. This phase of the research was an explorative one since our ambition was not to create a representative sample but rather to collect a relatively wide range of opinions.

Figure 1. Flowchart of investigations.

I do recognize that the authors think of this as an exploratory study, but I would like a bit more detail about the data. For example, are there measures of interviewee burden?  Cooperation?  Failure to complete?

Response

The research topics were considered very important and interesting questions by all participants, which is why the willingness to participate in the interviews was very high. People evaluated their participation in the research very positively and they were cooperative. Participants were willing to share their experiences and views with others on different problems related to the school catering system development.

It was a very frequent reflection that: ”There is so much talking about school feeding but this is the first time that my opinion is important for someone”. Or, as one parent formulated: “This is a very long-lasting problem and a crucial issue but lack of money and attention is an obstacle to improve the situation, so there is not too much to expect”.

The characteristic features of interviewees are summarised in Supplement 1.

Did your data analysis reveal anything surprising?  Since this was an exploratory study, what did you learn that you had not anticipated?

Response

There is an increasing number of municipalities that buy the catering service for schools from specific enterprises, which run the kitchen at schools or run the finishing kitchens, where they heat and serve the meals prepared in the central kitchens of the enterprises. The catering service provision has been considered a flourishing business. One anonymous source with a considerable level of experience in the private detective sphere has informed us that the different catering sector providers are his regular business partners because his firm is asked to uncover hidden microphones and voice transmitters in the offices and cars of the decision makers of these enterprises.

The catering service managers are the bosses of kitchens or finishing kitchens. In general, their work consists of the procuration of raw materials, food preparation and management of serving process. All respondents agreed that local managers are extremely important in the SCS but their scope of decision is rather limited due to budget limits and the difficult systems of regulations. A surprising phenomenon has occurred, namely the burnout the catering managers.

A catering manager said: “I am sick and tired of hearing from early morning to late night each day the snivelling spoiled children. They encourage each other to refuse to eat our food, and they complain about the bad meals we prepare.” The catering managers are frontline solders of the system; however, they felt abandoned. They complained in the following way: “From these limited material resources (i.e. a lack of money) we are not able to buy the raw materials for the kind of food which could satisfy the requirements of Decree… I do not know enough recipes to prepare a diverse menu to satisfy the requirements – we did not receive any help to do this.” One school canteen manager said: “The most important days for us are Mondays and Fridays: on Monday we have to prepare to energy-rich food, because the children wants nutrient rich food after the unsatisfactory food consumption in the week and, and on Friday we have to “fill up” the children with copious food”.

The role of parents is also critical: (1) there are parents who simply do not have energy/time to care about the food consumption of their children. There are those (2) who worry about the low quality/quantity of food served to their children in school canteens, which is why they pack them sandwiches or give them money to buy additional food, while (3) in the case of poor families the school canteen plays an important role in relieving families (and their budgets) from the burden of daily food provision. One school canteen manager said: “The most important days for us are Mondays and Fridays: on Monday we have to prepare to energy-rich food, because the children wants nutrient rich food after the unsatisfactory food consumption in the week and, and on Friday we have to “fill up” the children with copious food”.

The overburdened teachers are hardly suitable for cooperation. Teachers play a very important role in the formation of the eating behaviours and eating habits of children. The teachers eat mainly the same food as the children, if they are eating in the same canteen. Under these conditions there is an increasing tendency towards overburdened, burnt out teachers. As one teachers formulated it: ”I have a lot of problems in school and in my private life, I am simply too tired to deal with such problems as the nutrition of the children”. As another formulated it: “I am fed up with the fact that society tries to push all its problems onto the schools and teachers. I do not feel it to be my responsibility to care about what children eat under such conditions, when I have to teach them the most elementary rules of social behaviour, just because their parents are playing with their smartphones or lingering on social websites.”

I believe the thought about how children are key players here but have little input could be crucial. Can you develop that point a lot more? How can this type of analysis help do make that point?

Response

Our interviews highlighted that the current world of the SCS is quite distant from the demands of children. There is old, adolescent-sized furniture and sometimes rude and un-motivated kitchen staff (mainly older, often burned-out female employees), who acquired their experiences in years of relative food shortage and are not able to communicate in an appropriate way with children. Catering reform will only be successful if pupils like and choose to eat these meals, but this aspect has been neglected in Hungary.

The majority of specialists agreed that time has passed the old-style catering infrastructure by, and it has not been renovated for decades. This contrasts noticeably with the vivid colours and modern interiors of the majority of fast-food restaurants which target the younger generation. Under these conditions there is just a relatively low chance of attracting young consumers, who often consider traditional food as old-style.

The results of investigations have demonstrated clearly that children are key players in the Hungarian SCS but have little influence on the system. In Hungary, there does not exist any mechanism to survey the opinion and test the preference system of children. The most reliable indicators are the cleaners, who can furnish some information on the quantity of food and the amount leftover. There are neither experiences nor resources or mechanisms to uncover the drivers behind children’s behaviour [58].

I am a bit confused about Figure 1 (discussed lines 371-380). What precisely is “INFLUENCE”? DEPENDENCE?  I suspect influence is how well they can impact the food offerings in the schools, but dependence is not that straightforward.  Is dependence just a measure of how each actor/group relies upon the others in the scenario?  On line 376, you state that the influence of the BUSINES is much lower than the others, in the previously discussed 3-group, but clearly they have higher influence than teachers on the graph.

Response

The matrix of the direct influences of actors is shown in Table 4. This is a rectangular matrix. The main diagonal of the matrix (the influence of a given actor on itself) is by definition zero. The actor considered as the influencer is placed in the row, and the influenced party is situated in the column. The values in the cells of matrix are the simple averages of responses obtained as a result of discussion. To be on the safe side we have calculated the averages by group of actors (e.g. teachers, parents, catering managers etc., but we have not been able to prove significant differences between the two methods. The in-depth analysis of standard deviations according to interviewees could go beyond the limits of the current research.

Table 4. The matrix of direct influences on actors measured on a 0-4 scale.

GOV

  MUNICIP

PARENTS

CHILDREN

MANAGERS

BUSINESS

TEACHERS

Government GOV

0

3

1

0

1

4

2

Local authorities   MUNICIP

1

0

1

0

1

4

0

PARENTS

1

3

0

3

0

0

2

CHILDREN

0

0

2

0

0

0

1

Catering service   managers MANAGERS

0

0

0

2

0

0

1

Catering service   providers  BUSINESS

1

1

0

0

3

0

0

TEACHERS

1

2

2

3

1

1

0

Source: Authors’ own construction

Some remarks about the results of the matrix of direct influences based on in-depth interviews are indicated as follows:

-The matrix highlights the considerable influence of the government on the behaviour of catering service providers and on the SCS in general, because the government has a considerable influence on the monetary resources for school catering.

-Local authorities can exercise an important influence on catering service providers because they have the possibility to select the SCS provider for different schools in the framework of the public procurement procedure.

-In the opinion of interview partners a basically positive tendency has occurred over the last years – the increasing influence of parents’ organizations on the life of schools. At the same time, due to the lack of democratic roots and traditions it is quite difficult to achieve a situation in which parents’ influence is based on an absolute majority of the parents and not just on a relative majority of a small group with active and noisy members.

The bargaining position of different actors has been depicted on a two dimensional coordinate system. On one ordinate of the system we have indicated the level of influence of the given actor. This value has been calculated based on Table 4 by the summation and normalization of direct influences for each given actor in relation to another actor [37]. The dependence of actors have been calculated in a similar manner.

We agree with the reviewer’s comment because the owners of the SCS firms have approximately the same level of dependence as the former three actors, with a much lower influence.

 In the discussion of Table 6, you state (line 386) that good taste and children’s health have “the highest values,” but healthy food has a higher score than good taste, as does minimisation of expenditure on health promotion.  So this statement seems incorrect.  And then the analysis takes ”dimensionless” numbers and adds them together (lines 387-8) and compares them to other sums. Do they really work that way in some understandable sense? 

Response

Thank you. Correct. The rephrased sentence is as follows: Analysing the mobilizing force of the different interests good taste and children’s health have the highest value (Table 6). Notwithstanding, it is important to stress that the mobilizing force of the sum of cost minimization and simplicity is much higher than the healthiness of food criteria.

 I do not understand the axes of the 2 dimensions in Figure 2 (lines 393-395). Are they some kind of factor loadings or principal components?

Response

The indicator “mobilizing force” of different goals has been calculated based on the acceptance of differing goals weighted by bargaining power (influence) on actors. Analysing the mobilizing force of the different interests good taste and children’s health have the highest value (Table 6). Notwithstanding, it is important to stress that the mobilizing force of the sum of cost minimization and simplicity is much higher than the healthiness of food criteria.

Table 6. Mobilising force of different goals.

Goal

Mobilising   force

good taste of the meal

12.8

healthy food

17.1

healthy children

17.8

vote maximisation

8.6

feeling of being sated

10.2

minimisation of expenditure on   health promotion

17.4

simplicity of food preparation

2.4

Source: Authors’ own construction

In the Discussion, you highlight 5 key points (lines 413-423), yet your analysis only seemed to deal directly with the last one (line 422-3). 

Response

You are right. Our intention was to highlight and reaffirm the importance of school catering in health-related education. We did not elaborate the 5 key points further. The attitude of experts towards school catering has not been taken into consideration, neither in the choice of experts, nor in the interview phase.

This last issue brings up the key point about what is the precise contribution of this paper, and it was not clear to me what the paper did to change the landscape of how to think about making changes to school lunches to make them more healthful.  But I suspect that your analysis did do something, perhaps a lot, in that regard. It would be useful to make fairly specific statements about the contribution of this research.  Let us know explicitly (though briefly), the state of knowledge before your analysis, what your analysis adds to this, and what your analysis suggests are important next steps that should be taken to understand better the process of making successful changes. You highlight this as an exploratory study, so let us know explicitly what you found and how this will help make progress.

Overall, I am a bit confused about the goal of this paper.  Is it to look at some “exploratory” data and then make policy recommendations? Or is it to provide a better foundation for understanding the processes that need to happen for successful implementation of school lunch reforms. I suspect you are mixing the two together, but they really are separate goals and should be presented as such.

Response

The aim of this study is to analyse the causes of this failure, based on a conceptual framework of institutional economics and a strategic modelling of different institutes by examining the results of 72 interviews (33 experts, 26 parents and 13 teachers) conducted with representatives of different stakeholders. The results highlight the lack of preparation for the introduction of the new regulatory framework, as well as the inefficient communication between the different stakeholders. In order to support children in eating healthfully, a complex nutrition education program and continuous dialogue is needed between teachers, parents, catering staff and the government.

The results of investigations have demonstrated clearly that children are key players in the Hungarian SCS but have little influence on the system. In Hungary, there does not exist any mechanism to survey the opinion and test the preference system of children. The most reliable indicators are the cleaners, who can furnish some information on the quantity of food and the amount leftover. There are neither experiences nor resources or mechanisms to uncover the drivers behind children’s behaviour [58].

As a conclusion of this study it can be determined that the position and the system of interests of different actors must be taken into consideration before setting up SCS related regulations. Children play a central role because they will be able indirectly influence parents and teachers. From this it follows that considerable resources should be mobilized to understand the driving forces of children’s behaviour and their taste. On the other hand from a retrospective perspective it is obvious that the government wanted to improve the SCS “by force”, following top-down logic, with minimal mobilization of material and human resources on education and widening the possibilities of raw material procuration.

The SCS should be better integrated into the general education program. For example, the participation of children in food preparation and servicing could promote family-life education and a more harmonious division of work between the genders, too. In order to establish a successful and effective strategy, a continuous dialogue is needed between parents, health promotion experts, teachers and their organizations. It should be emphasized that there is no universal solution: the specific needs of children should be taken into consideration because a good school meal is an investment in the future.

Reviewer 2 Report

Summary:

This qualitative study is on a topic that should be of interest to journal readers given the importance of school nutrition to children in Hungary and other countries. However, enthusiasm for the paper is much diminished by its lack of clarity. Part of this may be due to English as a second language by the authors. I recommend a thorough review and revision by a native English-speaker or someone who is more proficient in the subtleties of the English language. In its current form it is not possible to accurately critique the methods and results – both are unclear. Additional concerns have to do with the conclusions made that appear to extend beyond the data collected. It may be possible to publish this study, but not without extensive revisions to improve clarity.

Major Issues

Introduction:

Please provide more information about the national survey of Hungarian SCS (reference 21) such as the year that data were collected, the number of schools included in the survey, how the schools and respondents were selected, who completed the surveys, and then specific data for all of the problems noted.  For example, sometimes a specific number is provided (e.g., 10% of schools have nutrition specialists involved in creating school menus), whereas other times not (e.g., lack of professional menu-planning). In addition, the meaning is unclear in some places (e.g., when you say that 2/3 of schools offer free access to safe drinking water in bathrooms, does that mean 1/3 don’t have any drinking water in bathrooms and what about elsewhere on the school property?  If ample drinking water is available elsewhere, is it important to have drinking water in the bathroom?).

It would also be helpful to provide more description of the Hungarian SCS in the introduction. Do all schools have an SCS? Do all receive government subsidies for meals?  What meals/snacks are provided?  What percent of children eat the SCS meals/snacks? Are foods brought from home more or less nutritious than what the schools provide? What were the existing standards prior to the 2015 updates? You state that the new rules were designed to increase whole grains, dairy, fruits and vegetables and decrease sodium, sugar and saturated fats, but how?  What were the specific requirements? This will help the reader understand the importance of the SCS and give helpful context for this study.

The study purpose is confusing:  “to uncover the causes of nutritional deficiencies of the SCM in Hungary”.  Do you mean the SCS?  And do you mean that you are trying to understand why the SCM was not implemented as intended?

Methods:

I found the phases of the study confusing. Why was the second phase of the study completed before the first phase? How can the 20 stakeholders involved in the first phase be depicted in the same Table 1 as the 24 stakeholders in the second phase?  I think what you are saying is that the stakeholders from phase 1 are from the same categories as in phase 2. If this is the case, it would be better to describe the types.

What was different about the 33 experts (line 157) versus the 18 experts (line 175) interviewed?  The purpose of each of the 4 sets of interviews is unclear. Consider a table illustrating who, how, why each of these sets of interviews was conducted.

You state that interviews in the first phase of the study were recorded, but do not provide any details about subsequent interviews. Given that this was done for the first phase, I was expecting the results to include some verbatim quotes from stakeholders. Were interviews in the second phase also recorded? If so, how? 

Do parents and teachers have direct experience with school meals?  It would be helpful to describe the nature of this direct experience. Do they eat the meals?  Or just hear about them from the children?  If the latter, I would call that indirect experience. Please justify and clarify the criteria you used to define ‘experts’ (line 139).

Results:

It is unusual to bring in findings from others in a results section (e.g., ‘in the opinion of Gaal et al). These sorts of statements are sprinkled throughout the results and should be moved to the discussion section instead.

What is the utility of Table 1 besides showing the number of participants in the second phase of the study? This information could be moved to the text. Please explain what a catering service entrepreneur is/does as well as the other participant categories.

There are two Tables numbered 2. What do the numbers represent in Table 2 (the first one)?  Please add total sample size to all tables. In the abstract you state that 83 interviews were conducted but when I try to follow the number of interviews from the various phases described in the methods, I get to: 20 (line 122) + 24 (line 132) + 33 (line 157) + 18 (line 175) which equals 95.

Tables 2 (second one) and 3 do not seem necessary. The longer versions of the words can be used in Table 4 or Table 5 or footnotes explaining the shorthand can be added to them. It is unclear where the numbers in Tables 4 and 5 came from. It appears from the supplementary materials that stakeholders were asked to rate influences and goals, so are these means from those ratings?  How does one interpret the direction of influence in Table 4? Where do the data for Table 6 come from and what do the numbers mean? What does ‘mobilizing force’ mean?

Similarly, it is unclear how Figures 1 and 2 were developed and what they mean. What does ‘convergence of interest’ signify?

Discussion:

The discussion does not follow from the results. Here one would expect to have the results put into context and the interpretation of the findings provided.

It is not at all clear how the conclusions numbered 1-5 were derived from the data collected.

In the abstract you state that the results highlight the lack of preparation for the introduction of the new regulations and inefficient communication between the different stakeholders. But it is unclear from the results sections how these statements are supported. The conclusions appear to extend beyond the data collected and presented.

Minor Issues

‘Focus groups’ is listed as a keyword, yet as far as I can tell, no focus groups were conducted; also I believe you mean that ‘prevention’ and ‘semi-quantitative methods’ as two distinct keywords.

In the introduction, it would be helpful to note which countries in Europe (if any) do not have any school nutrition guidelines.

Most of the last paragraph of the introduction is not necessary given that you are following the typical structure of a research article (i.e., introduction, methods, results, discussion).

Reference for the ‘WHO tool’ (line 426) is not provided.

Author Response

Reviewer 2

The reviewer’s useful comments and suggestions are highly appreciated. We have revised the manuscript in accordance with the reviewer’s comments. We have revised the manuscript in accordance with the reviewers’ comments. We have revised the Abstract, Introduction, Materials and Methods, Results, Discussion and conclusions sections, and References, rewritten many parts, and searched for and updated literature and provided additional information on the reform of school catering in Hungary.

Overall, we have made major revision to this manuscript. Firstly, the references have been updated and expanded. We carefully selected 62 papers, which are considered important or innovative studies, or comprehensive reviews offering us a wider picture of the reform of school catering in Hungary. In the new version of the manuscript, 13 references have been added. Secondly, the structure and sections have been revised, subsections have been added and several new paragraphs, tables and figures have been inserted. This version now includes revised Abstract, Introduction, Materials and Methods, Results, Discussion and conclusions sections, and References. Revised sections and subsections have been carefully reorganized, and new and modified paragraphs added to each section. Thirdly, language and grammar have been edited under the guidance of a professional native speaker.

Thank you very much for your comments and for taking the time to thoroughly review the manuscript. Your comments have been very constructive in helping us to clarify our message.

Summary:

This qualitative study is on a topic that should be of interest to journal readers given the importance of school nutrition to children in Hungary and other countries. However, enthusiasm for the paper is much diminished by its lack of clarity. Part of this may be due to English as a second language by the authors. I recommend a thorough review and revision by a native English-speaker or someone who is more proficient in the subtleties of the English language. In its current form it is not possible to accurately critique the methods and results – both are unclear. Additional concerns have to do with the conclusions made that appear to extend beyond the data collected. It may be possible to publish this study, but not without extensive revisions to improve clarity.

Major Issues

Introduction:

Please provide more information about the national survey of Hungarian SCS (reference 21) such as the year that data were collected, the number of schools included in the survey, how the schools and respondents were selected, who completed the surveys, and then specific data for all of the problems noted.  For example, sometimes a specific number is provided (e.g., 10% of schools have nutrition specialists involved in creating school menus), whereas other times not (e.g., lack of professional menu-planning). In addition, the meaning is unclear in some places (e.g., when you say that 2/3 of schools offer free access to safe drinking water in bathrooms, does that mean 1/3 don’t have any drinking water in bathrooms and what about elsewhere on the school property?  If ample drinking water is available elsewhere, is it important to have drinking water in the bathroom?).

It would also be helpful to provide more description of the Hungarian SCS in the introduction. Do all schools have an SCS? Do all receive government subsidies for meals?  What meals/snacks are provided?  What percent of children eat the SCS meals/snacks? Are foods brought from home more or less nutritious than what the schools provide? What were the existing standards prior to the 2015 updates? You state that the new rules were designed to increase whole grains, dairy, fruits and vegetables and decrease sodium, sugar and saturated fats, but how?  What were the specific requirements? This will help the reader understand the importance of the SCS and give helpful context for this study.

Response

Thank you for your valuable remarks and comments. We have considerably modified and extended the Introduction section.

The shool catering system (SCS) has a predominant place in the formation of nutrition behaviour [15-17] even if the efficiency of the SCS in the prevention of obesity has not been proven satisfactorily by rigorous, long-range studies [18, 19]. School lunch nutrition standards are the basis for improving the nutritional intake of all schoolchildren. All member states of the EU have policies to help schools to provide nutritionally balanced meals (EU, 2018). The SCS in selected member states of the EU is summarised in Table 1.

Table 1. School meal provision systems in selected member states of the EU.

Country

School meals are funded by the government

Lunch should follow national dietary   guidelines

Take up

Cost per meal

Nutrition education program (children/parents/staff)

or meal program

England

No

Yes

Average   FSM take up >60%

Average   meal £1.00-£2.00

NA

Finland

Yes

Yes

Average   take up >80%

NA

Yes

France

No

Yes

NA

Average   meal £2.50

NA

Germany

No

No

NA

NA

Yes

Italy

No

No

NA

Average   meal £2.50

Yes

Spain

No

No

NA

Average   meal £2.50

Yes

Sweden

Yes

Yes

Average   take up >80%

NA

NA

The Republic of Ireland

No

Yes

NA

Average   meal £1.00-£2.00

NA

Legend: FSM - free school meals, NA - Data are not available

Source: Authors’ own construction, based on Harper et al. [20]

It can be stated that in developed countries there is an increasing tendency to apply the SCS not just as a part of school logistics and the social system, but also as means of health and family-life education. That is why the importance of SCS has increased rapidly. The quality of the Hungarian SCS has been a highly debated issue for generations [21]. There are different business models: the majority of kitchens are owned by local authorities; however, several kitchens are run by catering service provider enterprises.

The Hungarian National Survey of SCS (called as Canteen Panorama) is a regular report of the National Institute of Pharmacy and Nutrition (former Hungarian Office of Food and Nutrition) [22-24]. The essential results of the last three surveys are summarised in Table 2.

Table 2. Summary of the surveys of school catering systems.

Characteristic features/indicators

Canteen panorama

2008

Canteen panorama

2013

Canteen panorama

2017

No. of schools (educational institutions)

3099

260 representatively chosen

139 elementary school

Elementary schools in the sample (%)

62

62

100

Secondary schools in the sample (%)

17

31

Elementary and secondary schools (%)

7

Boarding (%)*

92

: 17

100

School milk program (%)

15

35

72

Free fruits and vegetables (%)

14

78

95

Free drinking water outside the bathroom (%)

36

58

75

Proportion of students eating at the school canteen (in %)

Primary (7-10 years)

85

87

88

Lower secondary (%) (10-14 years)

47

63

61

Secondary school (%) (14-18 years)

20

27

Survey of children’s’ satisfaction conducted by the schools (%)

29

Qualified food catering manager (%)

76.5

83

no data

Food planning with involvement of a dietitian (%)

10

29

38

Energy and nutrients   content calculation (%)

22

57

no data

Net food budget   calculated for raw material**

0.72 EURO

1.2 EURO

1.15 EURO

School snack bar in   institutions (%)

45

50

44

Vending machines (%)

30

34

*In each elementary school, and proportion of secondary schools without boarding

**Hungarian currency converted to EURO on yearly average conversion rate

Source: Authors’ own construction, based on Martos [22] and Bakacs et al. [24], and Bakacs et al. [23]

The Hungarian SCS is an evolving system, which is characterised by considerable backwardness and delayed development, compared to Western-European school boarding solutions. The national averages hide important regional differences. According to the Canteen Panoramas [22-24], the school snack bars and vending machines are significant competitors for the SCS, and have an improving product portfolio. At the same time the SCS is not capable of meeting changing demands. A good indicator of this is the rapidly decreasing take up rate as children’s age increases. Another indicator of problems is the fact that according to the school canteen survey (Bakacs et al. [24]), 85% of parents prepared some kind of prepacked food for their children in 2013. Prepacked food was mainly sandwiches, with vegetables (41%) or without vegetables (37%), and refreshing soft drinks (50%).

Generally speaking, the Hungarian SCS faces substantial long-standing, unsolved problems, which can be attributed to the lack of monetary resources and the lack of attention from responsible organs. This motivated the government to take action to update the nutrition requirements of the public catering service, including school meals, in 2011. In 2011 the Office of the Hungarian Chief Medical Officer issued the ‘Recommendation for Public Caterers’ with nutritional standards [25]. This recommendation provided a checklist enabling to monitor the adherence to the recommendations. This document was the basis of the 37/2014 decree of Ministry of Human Capacities (MHC) on public catering [26].

The aim of the school meal provision was to reduce the prevalence of obesity and non-communicable diseases (NCDs) among Hungarian children and adolescents, as well as promote healthier environments, especially in schools. The most important elements of the decree are summarised in Table 3. This rather ambitious regulation has tried to increase the fruit, vegetable, cereals and milk consumption and decrease the consumption of fat, sugar and salt. When comparing the content of the Decree with dietary guidelines of other member states of the EU, our nation's dietary guidelines seem to be in line with the nutrition policy of most EU member states. However, the public acceptance of this new regulation has been mixed, and mainly negative. Overall take up rates were generally low, according to the comments of school children. Pupils refused the dishes which conformed to the requirements and both children and their parents rebelled against the rules. In 2015 the Hungarian Association of Dietitians carried out a survey among dietitian food-service managers about the practical feasibility of Hungarian Regulation No. 37/2014 on nutrition requirements in the provision of public food services. Of the 56 food-service managers interviewed 19 represented child nutrition institutions. Since the introduction of the regulation in 36 of 56 institutions interviewed satisfaction with nutrition care had decreased. In 13 cases the rate of dissatisfaction was 30% or more, and the amount of daily food waste increased significantly. The majority of catering service providers (62%) requested some alterations to the regulations because the prescribed composition of the food was not in line with children’s demands. The greatest cause of dissatisfaction among parents and children derived from the control of salt content, and the attempt to provide the prescribed quantity of dairy products and added sugars [27]. To date, there are no representative, academically well founded empirical studies on pupils’ consumption of the new school meals [28]. That is why, in 2016, the regulatory framework was changed [29]. The new decree significantly modified the prescriptions. The most important features of the original and the modified decree are summarised in Table 3.

Table 3. The most important changes in the public catering decrees.

2014*

2016**

Food-based standards

Food-based standards

              Specific foods and food groups   have to be provided daily for all age groups

                       (for one person)

5 meals/day

4 portions of fruits or   vegetables per day, at least one of which should be raw

4 portions of fruits or   vegetables per day, at least one of which should be raw

3 portions of cereals, at   least one which should be whole grain

3 portions of cereals, at   least one which should be whole grain

0.5 l milk or a diary product   with an adequate amount of calcium

-

Nursery (1 to <3 years)

3 portions of fruits or   vegetables per day, at least one of which should be raw

3 portions of fruits or   vegetables per day, at least one of which should be raw

2 portions of cereals, at   least one of which should be whole grain

2 portions of cereals, at   least one of which should be whole grain

0.4 l milk or a diary product   with an adequate amount of calcium

-

3 meals/day

2 portions of fruits or   vegetables per day, at least one of which should be raw

2 portions of fruits or   vegetables per day, at least one of which should be raw

2 portions of cereals, at   least one of which should be whole grain

2 portions of cereals, at   least one of which should be whole grain

0.3 l milk or a diary product   with an adequate amount of calcium

-

1 meals/day

1 portion of fruits or   vegetables per day, at least three of which should be raw over a 10-day   catering period

1 portion of fruits or   vegetables per day, at least three of which should be raw over a 10-day   catering period

Supplementation

If pre-primary, 5 or 3 meals   are provided a day, milk or a diary product with an adequate amount of   calcium should be served every day

Regulations, limitations and   prohibitions of using certain foods and products

Fat content of milk

-2.8% or 3.6% milk fat milks   should be served for age group 1-3

-1.5% or <1.5% milk fat   milks should be served above 3 years old

-2.8% or 3.6% milk fat milks   should be served for age group 1-3

-2.8% or <2.8% milk fat   milks should be served above 3 years old

Water

Constant access to fresh water   (outside of bathrooms)

Constant access to fresh water   (outside of bathrooms)

Added sugar

Free sugar should not exceed   8% of total energy in a 10-day catering period

Free sugar should not exceed   10% of total energy in a 10-day catering period

Salt and free sugar

Salt and sugar should not be   placed on dining table.

Salt or sugar storers should   be labelled: "Excessive salt intake could cause cardiovascular diseases,   obesity and diabetes! "

*Public Catering Decree EMMI (Ministry of Human Capacities), Decree 37/2014 (came into force in September 2015)

** Decree on modification of EMMI (Ministry of Human Capacities), Decree 37/2014 (came into force in December 2016)

Source: MHC [26] and MHC [29]

The Hungarian SCS is a highly complex, dynamic system. The last Decree on Public Catering has yielded important results: the quality of meals has been increasing in the last few years [30]; however, numerous unresolved problems have remained. The new regulatory framework of the SCS is more liberal, although in numerous points it represents a step back from the goals of the former Decree. The aim of this study is to uncover the causes of this phenomenon. In a wider context our goal is to understand the justification of the low acceptance of the SCS towards a regulatory attempt to manage a long-standing, generally accepted problem, namely childhood obesity. What is the main reason that the regulation could not be put into practice? How can we explain that the take up rate of the SCS has not increased considerably since the introduction of the new Decree. To achieve this goal the authors have applied an innovative, semi quantitative method for the analysis of social bargaining.

The study purpose is confusing:  “to uncover the causes of nutritional deficiencies of the SCM in Hungary”.  Do you mean the SCS?  And do you mean that you are trying to understand why the SCM was not implemented as intended?

Response

Correct. Thank you. We have corrected the abreviation of the shool catering system (SCS).

Methods:

I found the phases of the study confusing. Why was the second phase of the study completed before the first phase? How can the 20 stakeholders involved in the first phase be depicted in the same Table 1 as the 24 stakeholders in the second phase?  I think what you are saying is that the stakeholders from phase 1 are from the same categories as in phase 2. If this is the case, it would be better to describe the types.

Response

Thank you so much for your comments. This section has been restructured.

2.1. The methodological framework of the research

The fundamental theoretical paradigms of the analysis were institutional economic theory [31, 32], principle-agent theory [33] and the concept of strategic planning [34]. According to the basic theory of the so-called “French school of strategy” the different social systems can be considered as a playground in which different groups of participants (the actors) take part with the purpose of making their specific interests prevail. In the opinion of [35], if one can adequately simplify the actors and the most characteristic features of their systems of interests and strategies, then it is possible to analyse the chances of different actors realizing their goals. The method of the systematic analysis of social bargaining can be described by using the MACTOR model. One of the key concepts of the model is that actors may influence other actors in terms of their potential to put pressure on other actors directly or indirectly in order to affect their behaviour. The influence of one actor (A) on another actor (C) is the sum of the direct and indirect influences of actor A on actor C.

Based on unstructured interviews, the key actors of the catering system were determined. In the next phase the intensity of mutual direct influences was characterized using a rectangular matrix offering a good overview of the MACTOR method. The cells of the matrix – by definition – reflect the intensity of the influence of any actor in a row on any actor in a column. The intensity of the direct influence by one actor on another was measured on a 0-4 scale ranging from no influence to absolute influence.

The importance of different goals from the point of view of each actor was expressed by the Matrix of Actor-Objective. This was the so-called 1MAO matrix. Each cell of the matrix contained the attitude of a given actor towards a given goal in the form of a positive, 0 or negative sign. In the second phase the 2MAO matrix is determined, which contains the intensity of these attitudes determined from the point of view of different actors and quantified on a -4 …+4 scale, where -4 denotes the high importance and total negation of the given goal, and +4 denotes the high importance and total support.

The mathematical methodology of the MACTOR method is presented in the literature [36].

2.2. Setting up the input data system

The data collection for the analysis was a multiphase process.

In the present study we applied a self-designed interview method. Besides analysis of publicly available papers, press releases, newspaper articles and the blogosphere, face-to-face expert estimations were made with 24 stakeholders related to the field of Hungarian catering. This series of preliminary interviews were conducted with the purpose of determining the set of relevant actors and interests. The interviews were carried out in 2015 and 2016. The aim of this preliminary phase of interviews was to outline the most important stakeholder groups and the set of the potential objectives of the stakeholders. As a result of these preliminary investigations a robust and relatively well-manageable set of actors and goals could be identified. In setting up a pool of experts a specific procedure was followed. In this phase we pursued the following logic. We considered experts to be people (1) who have a direct “field” experiences in catering functions as parents or teachers; (2) people whose job directly involves a catering business with relatively long experience in the practice of SC and whose existence directly depends on this enterprise S; (3) independent experts, preferably those who have been especially active in professional social debates concerning the catering regulations in the printed and electronic media; (4) experts who have been actively involved in the preparation and enforcement of the new regulatory framework of the SCS. The attitude of experts towards school catering has not been taken into consideration, neither in the choice of experts, nor in the interview phase.

The second phase of research was a semi-quantitative interview. The list of potential participants was collected on the basis of intensive research into publications (including professional conferences, various formal and informal meetings of professional communities, the blogosphere and the grey literature), membership of professional organizations and the personal recommendations of other experts.

In summary, the names of 321 experts were collected (not including parents and teachers). Out of these experts we tried to make contact with specialists who supposedly – in the opinion of at least two members of the community of authors – have a more ‘holistic’ approach to the SCS universe without taking into consideration their attitude to the SCS question. In this way 78 experts were selected. We contacted 61 of them; 45 respondents expressed their willingness to participate in the research. Due to time and financial constraints 33 expert-interviews were carried out, all of them face-to-face. Additionally, we interviewed 26 parents and 13 teachers. The characteristic features of interviewees re summarised in Supplement 1.

The quantification of the intensity of actor-actor influences, as well as the actor-objective relations has been developed in a step-by-step manner. As we have experienced with our previous research [37], filling out the input matrices in the form of MS Excel worksheets for research was a very time-consuming (and in some cases a rather boring) process often leading to internal contradictions because it was very difficult to achieve a general common interpretation of different scales. That is why a semi-structured interview was used [38]. The conversion of the verbal estimations was carried out in the personal interview phase with the help of the researchers. The only task of the researchers was to help to interpret the different scales. This method provided to be an efficient method for achieving internal consistency in the input data for analysis [39].

In the framework of the interviews we asked the respondents to evaluate the bargaining power of each actor in comparison with another actor (e.g. Government vs. teachers, Government vs. Catering service managers etc…) on a 0-4 scale. The interpretation of this scale was the following:

0 – no direct influence

1 – actor A can eliminate the tactical steps of actor B

2 – actor A can jeopardize/eliminate the projects of actor B

3 – actor A can jeopardize/eliminate the strategic goals of actor B

4 – actor A can substantially influence/dominate actor B

In the second phase of interviews we asked the interviewees to evaluate the attitudes of actors (stakeholders) towards different elements of goal set on a -4… 0 … +4 scale. The interpretation of the scale was as follows:

-4 the objective is against the vital interest/jeopardizes the existence of the actor

-3 the objective jeopardizes the strategic mission of actors

-2 the objective jeopardizes the tactical goals of the actors

-1 the objective jeopardizes the operative goals of the actor

0 the actors’ attitude towards the goal is neutral

+1 the objective falls in line with the operative goals of the actor

-2 the objective falls in line with the tactical goals of the actors

3 the objective considerably supports the strategic goals of the actor

4 the objective is a vital interest of the actor

The participants received the cumulated input–matrices and their interpretation by e-mail, and had the opportunity to suggest some modifications. The results of the MACTOR analysis were discussed in detail with a representative pool of respondents in a group discussion and in face-to-face interviews. This phase of the research was an explorative one since our ambition was not to create a representative sample but rather to collect a relatively wide range of opinions.

Figure 1. Flowchart of investigations.

Source: Authors’ own construction

The research topics were considered very important and interesting questions by all participants, which is why the willingness to participate in the interviews was very high. People evaluated their participation in the research very positively and they were cooperative. Participants were willing to share their experiences and views with others on different problems related to the school catering system development.

It was a very frequent reflection that: ”There is so much talking about school feeding but this is the first time that my opinion is important for someone”. Or, as one parent formulated: “This is a very long-lasting problem and a crucial issue but lack of money and attention is an obstacle to improve the situation, so there is not too much to expect”.

2.3. Ethical issues

The Inter-Faculty Research Ethics Committee of the faculty of Budapest Corvinus University approved both the concept and procedure of the research (Ref. No.15/12/2014). All of the participants signed an informed consent, which described the procedure of the research in detail.

Supplementary materials

Table 1. Structure of participants in preliminary phase of research.

Preliminary   interview phase

no. of participants

%

Catering   service entrepreneur

2

8.33

Catering   service manager (dietician)

3

12.50

Teachers

5

20.83

Independent   nutritional specialists

4

16.67

Government

1

4.17

Local   authority

2

8.33

Parents

7

29.17

Total

24

100

Source: Authors’ own construction

Table 2. The socio-economic characteristic features of participants.

Highest level of qualification

Gender

Age

Place of residence

High school

BSC

MSc or higher

male

female

<35

36-50

>50

Budapest (capital)

Big town

Small town

Village

Catering service entrepreneur

1

3

2

3

3

1

4

1

2

1

1

2

Catering service manager

2

8

2

1

11

2

8

2

6

1

3

2

Independent nutritional   specialists

6

6

3

2

1

6

Government

5

2

3

1

2

2

5

Local authority

1

3

2

2

1

3

1

1

1

1

Parents

13

9

4

5

21

13

12

1

12

4

5

5

Teachers

9

4

1

12

2

7

4

3

4

3

3

Source: Authors’ own construction

What was different about the 33 experts (line 157) versus the 18 experts (line 175) interviewed?  The purpose of each of the 4 sets of interviews is unclear. Consider a table illustrating who, how, why each of these sets of interviews was conducted.

You state that interviews in the first phase of the study were recorded, but do not provide any details about subsequent interviews. Given that this was done for the first phase, I was expecting the results to include some verbatim quotes from stakeholders. Were interviews in the second phase also recorded? If so, how? 

Do parents and teachers have direct experience with school meals?  It would be helpful to describe the nature of this direct experience. Do they eat the meals?  Or just hear about them from the children?  If the latter, I would call that indirect experience. Please justify and clarify the criteria you used to define ‘experts’ (line 139).

Response

The data collection for the analysis was a multiphase process.

In the present study we applied a self-designed interview method. Besides analysis of publicly available papers, press releases, newspaper articles and the blogosphere, face-to-face expert estimations were made with 24 stakeholders related to the field of Hungarian catering. This series of preliminary interviews were conducted with the purpose of determining the set of relevant actors and interests. The interviews were carried out in 2015 and 2016. The aim of this preliminary phase of interviews was to outline the most important stakeholder groups and the set of the potential objectives of the stakeholders. As a result of these preliminary investigations a robust and relatively well-manageable set of actors and goals could be identified. In setting up a pool of experts a specific procedure was followed. In this phase we pursued the following logic. We considered experts to be people (1) who have a direct “field” experiences in catering functions as parents or teachers; (2) people whose job directly involves a catering business with relatively long experience in the practice of SC and whose existence directly depends on this enterprise S; (3) independent experts, preferably those who have been especially active in professional social debates concerning the catering regulations in the printed and electronic media; (4) experts who have been actively involved in the preparation and enforcement of the new regulatory framework of the SCS. The attitude of experts towards school catering has not been taken into consideration, neither in the choice of experts, nor in the interview phase.

The second phase of research was a semi-quantitative interview. The list of potential participants was collected on the basis of intensive research into publications (including professional conferences, various formal and informal meetings of professional communities, the blogosphere and the grey literature), membership of professional organizations and the personal recommendations of other experts.

In summary, the names of 321 experts were collected (not including parents and teachers). Out of these experts we tried to make contact with specialists who supposedly – in the opinion of at least two members of the community of authors – have a more ‘holistic’ approach to the SCS universe without taking into consideration their attitude to the SCS question. In this way 78 experts were selected. We contacted 61 of them; 45 respondents expressed their willingness to participate in the research. Due to time and financial constraints 33 expert-interviews were carried out, all of them face-to-face. Additionally, we interviewed 26 parents and 13 teachers. The characteristic features of interviewees re summarised in Supplement 1.

The quantification of the intensity of actor-actor influences, as well as the actor-objective relations has been developed in a step-by-step manner. As we have experienced with our previous research [37], filling out the input matrices in the form of MS Excel worksheets for research was a very time-consuming (and in some cases a rather boring) process often leading to internal contradictions because it was very difficult to achieve a general common interpretation of different scales. That is why a semi-structured interview was used [38]. The conversion of the verbal estimations was carried out in the personal interview phase with the help of the researchers. The only task of the researchers was to help to interpret the different scales. This method provided to be an efficient method for achieving internal consistency in the input data for analysis [39].

Results:

It is unusual to bring in findings from others in a results section (e.g., ‘in the opinion of Gaal et al). These sorts of statements are sprinkled throughout the results and should be moved to the discussion section instead.

Response

Thank you. Right. We have moved findings in to the results section.

What is the utility of Table 1 besides showing the number of participants in the second phase of the study? This information could be moved to the text. Please explain what a catering service entrepreneur is/does as well as the other participant categories.

Response

Thank you for that comments. There is an increasing number of municipalities that buy the catering service for schools from specific enterprises, which run the kitchen at schools or run the finishing kitchens, where they heat and serve the meals prepared in the central kitchens of the enterprises. The catering service managers are the bosses of kitchens or finishing kitchens. In general, their work consists of the procuration of raw materials, food preparation and management of serving process.

There are two Tables numbered 2. What do the numbers represent in Table 2 (the first one)?  Please add total sample size to all tables. In the abstract you state that 83 interviews were conducted but when I try to follow the number of interviews from the various phases described in the methods, I get to: 20 (line 122) + 24 (line 132) + 33 (line 157) + 18 (line 175) which equals 95.

Tables 2 (second one) and 3 do not seem necessary. The longer versions of the words can be used in Table 4 or Table 5 or footnotes explaining the shorthand can be added to them. It is unclear where the numbers in Tables 4 and 5 came from. It appears from the supplementary materials that stakeholders were asked to rate influences and goals, so are these means from those ratings?  How does one interpret the direction of influence in Table 4? Where do the data for Table 6 come from and what do the numbers mean? What does ‘mobilizing force’ mean?

Response

You are correct. The Materials and Methods section has been restructured to answer your remarks (see response to Methods).

Similarly, it is unclear how Figures 1 and 2 were developed and what they mean. What does ‘convergence of interest’ signify?

Response

Based on the actor-goal matrix (Table 5) we have depicted the position of different actors by the analysis of correspondence [42]. This method is widely applied for the visualization of the relative position of different actors to each other. The results of correspondence analysis are presented in Figure 3. Analysing the convergence of of interests (similarities of the interest system of different actors), four typical groups can be identified: the local authorities; the cluster of teachers, government and parents; the children, and the owners and managers of catering businesses.

Figure 3. Two–dimensional map of convergence between actors.

Source: Authors’ own construction

Discussion:

The discussion does not follow from the results. Here one would expect to have the results put into context and the interpretation of the findings provided.

Response

Thank you for these comments. The results and discussion sections have been restructured accordingly. Our result are in line with conclusions drown by another actors. For example, in the opinion of Gaál et al. [43] the central government of Hungary has almost exclusive power to formulate and realize strategic decisions and shape the regulatory framework of the health care system, as well as make public-health related interventions. The majority of schools (with the exception of schools owned by foundations and churches) are in state ownership.Due to a wide agreement on the long-term instability of the Hungarian health-care system [44-47], preventive measures should be given priority.

It is a contradiction that while all Hungarian governmental programs [48-51] have highlighted the importance of prevention in the health-care system, obesity among the young population groups has been increasing in the last few decades [52, 53].

It should be highlighted that neither schools, teachers, nor parents can be considered homogenous groups [54]. There are significant differences in parents’ attitudes and behaviour in relation to the school catering system. The results of van Zenten [55] and [56] highlight that upper-or middle-class parents have a much higher level of aspiration to participate in the decision–making process in the framework of the school than working–class parents, who are more ‘loyal’ to local schools [57].

The importance of school feeding for parents was a surprising result. It can be regarded as positive that the healthiness of school feeding has been evaluated as a question of great interest. This can be considered a favourable tendency.

The results of investigations have demonstrated clearly that children are key players in the Hungarian SCS but have little influence on the system. In Hungary, there does not exist any mechanism to survey the opinion and test the preference system of children. The most reliable indicators are the cleaners, who can furnish some information on the quantity of food and the amount leftover. There are neither experiences nor resources or mechanisms to uncover the drivers behind children’s behaviour [58].

The average take up level is low, varying from 40% to 95%; the highest take up rate in Europe is in Finland (over 90%), probably because school lunch plays a central role in education [20].

Based on our analyses, the failure of the new regulation of the SCS was predictable because children have not accepted dishes with low salt content and milk with low fat content. The take up level of the reform was extremely low as children were not involved in the new and rapidly introduced changes. The reluctance of children induced a chain reaction: neither the parents nor the teachers were motivated, nor did they have any background knowledge to argue for the positive aspects of the reform. The catering service providers and the local managers experienced a high level of food waste. These negative views were reinforced by the media, which over-emphasized the negative aspects of the new regulations. Under these conditions it was relatively easy to forge a coalition between parents, catering service providers and managers, teachers and local authorities to choose the easier option: i.e. to force the government to substantially modify the original legislation. The fate of this regulation, based on sound professional arguments, lends itself to an analysis of the failures leading to the collapse of the regulation. The most important failures can be summarized as follows:

1.        There was no general, well designed study and testing of the effects of the regulation in the framework of a pilot-study.

2.        There was no multilateral communication between the government and the stakeholders, mainly with catering service providers and catering managers.

3.        The teachers did not have the necessary background to discuss the issue with children due to their total illiteracy in the field of nutrition.

4.        Children do not have to learn practically any nutrition or food skills during their studies in the primary and secondary school so they did not have the capacity to understand the reason for the changes.

5.        The energy and resources of different governmental organs were scattered among different – partially competing – goals.

The school lunch provides an excellent opportunity to learn healthy eating habits, promote nutrient-rich foods, involve children in foodservice planning, and improve the nutritional intake of school children. There is an urgent need to improve the situation, taking into account the WHO tool for the development of school nutrition programmes in the European Region [59]. The policy paper emphasizes that healthy food and nutrition should be given a high priority on every school agenda and school meals are an indispensable part of a whole school approach to health promotion. To ensure effective implementation, stakeholders should review the available information on nutritional status and eating patterns before developing school lunch standards and issuing any regulation on nutrition requirements in public food-services.

As a conclusion of this study it can be determined that the position and the system of interests of different actors must be taken into consideration before setting up SCS related regulations. Children play a central role because they will be able indirectly influence parents and teachers. From this it follows that considerable resources should be mobilized to understand the driving forces of children’s behaviour and their taste. On the other hand from a retrospective perspective it is obvious that the government wanted to improve the SCS “by force”, following top-down logic, with minimal mobilization of material and human resources on education and widening the possibilities of raw material procuration.

Cooperation among all the different stakeholders is crucial when working out a school food and nutrition policy [60]. Moreover, one can even state that cooperation between different players, including coopetition, is a fundamental issue in today’s world [61, 62]. Representatives of teachers, parents, pupils, caterers, and representatives from a school’s governing body should develop an action plan and introduce a comprehensive education program. The SCS should be better integrated into the general education program. For example, the participation of children in food preparation and servicing could promote family-life education and a more harmonious division of work between the genders, too. In order to establish a successful and effective strategy, a continuous dialogue is needed between parents, health promotion experts, teachers and their organizations. It should be emphasized that there is no universal solution: the specific needs of children should be taken into consideration because a good school meal is an investment in the future.

It is not at all clear how the conclusions numbered 1-5 were derived from the data collected.

Response

You are right. Our intention was to highlight and reaffirm the importance of school catering in health-related education. We did not elaborate the 5 key points further.

In the abstract you state that the results highlight the lack of preparation for the introduction of the new regulations and inefficient communication between the different stakeholders. But it is unclear from the results sections how these statements are supported. The conclusions appear to extend beyond the data collected and presented.

Response

As a conclusion of this study it can be determined that the position and the system of interests of different actors must be taken into consideration before setting up SCS related regulations. The children play a central role because they will be able indirectly influence the parents and the teachers. From this follows that considerable resource should be mobilized to understand the driving forces of children behaviour and their taste. On the other hand from a retrospective perspective it is obvious that the government wanted to improve the SCS “by force”, following the top-down logic, with minimal mobilization of material and human resources on education and widening the possibilities of raw material procuration.

Minor Issues

‘Focus groups’ is listed as a keyword, yet as far as I can tell, no focus groups were conducted; also I believe you mean that ‘prevention’ and ‘semi-quantitative methods’ as two distinct keywords.

Response

Right. Keywords have been changed.

In the introduction, it would be helpful to note which countries in Europe (if any) do not have any school nutrition guidelines.

Response

The shool catering system (SCS) has a predominant place in the formation of nutrition behaviour [15-17] even if the efficiency of the SCS in the prevention of obesity has not been proven satisfactorily by rigorous, long-range studies [18, 19]. School lunch nutrition standards are the basis for improving the nutritional intake of all schoolchildren. All member states of the EU have policies to help schools to provide nutritionally balanced meals (EU, 2018). The SCS in selected member states of the EU is summarised in Table 1.

Table 1. School meal provision systems in selected member states of the EU.

Country

School meals are funded by the government

Lunch should follow national dietary   guidelines

Take up

Cost per meal

Nutrition education program (children/parents/staff)

or meal program

England

No

Yes

Average   FSM take up >60%

Average   meal £1.00-£2.00

NA

Finland

Yes

Yes

Average   take up >80%

NA

Yes

France

No

Yes

NA

Average   meal £2.50

NA

Germany

No

No

NA

NA

Yes

Italy

No

No

NA

Average   meal £2.50

Yes

Spain

No

No

NA

Average   meal £2.50

Yes

Sweden

Yes

Yes

Average   take up >80%

NA

NA

The Republic of Ireland

No

Yes

NA

Average   meal £1.00-£2.00

NA

Legend: FSM - free school meals, NA - Data are not available

Source: Authors’ own construction, based on Harper et al. [20]

It can be stated that in developed countries there is an increasing tendency to apply the SCS not just as a part of school logistics and the social system, but also as means of health and family-life education. That is why the importance of SCS has increased rapidly.

Reference for the ‘WHO tool’ (line 426) is not provided.

Response

Correct. Reference has been inserted.

Most of the last paragraph of the introduction is not necessary given that you are following the typical structure of a research article (i.e., introduction, methods, results, discussion).

Response

We have modified the manuscript following the structure of a research article.

Author Response

Reviewer 3

The reviewer’s useful comments and suggestions are highly appreciated. We have revised the manuscript in accordance with the reviewer’s comments. We have revised the manuscript in accordance with the reviewers’ comments. We have revised the Abstract, Introduction, Materials and Methods, Results, Discussion and conclusions sections, and References, rewritten many parts, and searched for and updated literature and provided additional information on the reform of school catering in Hungary.

Overall, we have made major revision to this manuscript. Firstly, the references have been updated and expanded. We carefully selected 62 papers, which are considered important or innovative studies, or comprehensive reviews offering us a wider picture of the reform of school catering in Hungary. In the new version of the manuscript, 13 references have been added. Secondly, the structure and sections have been revised, subsections have been added and several new paragraphs, tables and figures have been inserted. This version now includes revised Abstract, Introduction, Materials and Methods, Results, Discussion and conclusions sections, and References. Revised sections and subsections have been carefully reorganized, and new and modified paragraphs added to each section. Thirdly, language and grammar have been edited under the guidance of a professional native speaker.

Thank you very much for your comments and for taking the time to thoroughly review the manuscript. Your comments have been very constructive in helping us to clarify our message.

Feedback on: The reform of school catering in Hungary: Anatomy of a health-education failure School food and nutrition policies in various parts of the world encounter challenges with their development, implementation, and monitoring.

If food intakes among school students are to improve, jurisdictions need an improved understanding of how to increase the effectiveness of such policies.

This article, which examines policy (non)implementation in Hungary, makes a useful contribution by using an innovative stakeholder analysis. Currently, however, the article suffers from a number of serious weaknesses that need revision.

Include articles from Hungary on school food On line 83, the authors cite an odd reference on bankruptcy prevention when they indicate there are no empirical studies on pupils’ consumption of the new lunches. I am not from Hungary and know little about the country, however, I encourage the authors to read and include pertinent literature. A report on the Public Catering Decree in Hungary by Biro and Zental published by the World Health Organization indicates that positive change occurred in schools between 2013 and 2017 and the results of this assessment are examined in more detail in a recent article by Varga et al (Assessment of the public catering act in primary schools in Hungary). Another couple of articles, which may be less useful, are by Molnar et al (Current challenges faced by public catering . . . challenges is spelled challenges in the title of the copy I have) and an article by Toth et al (Improving knowledge, technology and food safety in school catering system in Hungary).

Response

Thank you very much for your valuable comments. We have substantially restructured and extended the Introduction section and inserted all suggested references with the exception of Molnar et al. and Toth et al. focussing mainly on food safety aspects of the topic.

Parallel with increasing awareness of the adverse effects of non-communicable diseases, e.g. the effects of obesity and obesity related diseases on the health condition and mortality of the population [1-3], the inappropriate diets and eating habits of children and adolescents is a much debated problem all over Europe [4-6]. Growing anxiety concerning the proliferation of the unhealthy eating habits of new generations [7], as well as the increasing number of overweight and obese children [8-10] has increased attention towards different methods of influencing children's and adolescents' health behaviour [11, 12]. This is an especially important problem in Hungary where the obesity rate is high. The overall prevalence of overweight and obesity among children and adolescents is 40% and 32%, respectively, and in women overweight and obesity are both at 32% [13]. This causes a considerable burden for the social security and health care system [14, 13]. Obesity has been described by the WHO as “a global epidemic” due to its high prevalence.

The shool catering system (SCS) has a predominant place in the formation of nutrition behaviour [15-17] even if the efficiency of the SCS in the prevention of obesity has not been proven satisfactorily by rigorous, long-range studies [18, 19]. School lunch nutrition standards are the basis for improving the nutritional intake of all schoolchildren. All member states of the EU have policies to help schools to provide nutritionally balanced meals (EU, 2018). The SCS in selected member states of the EU is summarised in Table 1.

Table 1. School meal provision systems in selected member states of the EU.

Country

School   meals are funded by the government

Lunch   should follow national dietary guidelines

Take   up

Cost   per meal

Nutrition   education program (children/parents/staff)

or   meal program

England

No

Yes

Average FSM take up >60%

Average meal £1.00-£2.00

NA

Finland

Yes

Yes

Average take up >80%

NA

Yes

France

No

Yes

NA

Average meal £2.50

NA

Germany

No

No

NA

NA

Yes

Italy

No

No

NA

Average meal £2.50

Yes

Spain

No

No

NA

Average meal £2.50

Yes

Sweden

Yes

Yes

Average take up >80%

NA

NA

The Republic of Ireland

No

Yes

NA

Average meal £1.00-£2.00

NA

Legend: FSM - free school meals, NA - Data are not available

Source: Authors’ own construction, based on Harper et al. [20]

It can be stated that in developed countries there is an increasing tendency to apply the SCS not just as a part of school logistics and the social system, but also as means of health and family-life education. That is why the importance of SCS has increased rapidly. The quality of the Hungarian SCS has been a highly debated issue for generations [21]. There are different business models: the majority of kitchens are owned by local authorities; however, several kitchens are run by catering service provider enterprises.

The Hungarian National Survey of SCS (called as Canteen Panorama) is a regular report of the National Institute of Pharmacy and Nutrition (former Hungarian Office of Food and Nutrition) [22-24]. The essential results of the last three surveys are summarised in Table 2.

Table 2. Summary of the surveys of school catering systems.

Characteristic features/indicators

Canteen panorama

2008

Canteen panorama

2013

Canteen panorama

2017

No. of schools (educational   institutions)

3099

260 representatively chosen

139 elementary school

Elementary schools in the   sample (%)

62

62

100

Secondary schools in the   sample (%)

17

31

Elementary and secondary   schools (%)

7

Boarding (%)*

92

: 17

100

School milk program (%)

15

35

72

Free fruits and vegetables (%)

14

78

95

Free drinking water outside   the bathroom (%)

36

58

75

Proportion of students eating   at the school canteen (in %)

Primary (7-10 years)

85

87

88

Lower secondary (%) (10-14   years)

47

63

61

Secondary school (%) (14-18   years)

20

27

Survey of children’s’   satisfaction conducted by the schools (%)

29

Qualified food catering   manager (%)

76.5

83

no data

Food planning with involvement   of a dietitian (%)

10

29

38

Energy   and nutrients content calculation (%)

22

57

no data

Net   food budget calculated for raw material**

0.72 EURO

1.2 EURO

1.15 EURO

School   snack bar in institutions (%)

45

50

44

Vending   machines (%)

30

34

*In each elementary school, and proportion of secondary schools without boarding

**Hungarian currency converted to EURO on yearly average conversion rate

Source: Authors’ own construction, based on Martos [22] and Bakacs et al. [24], and Bakacs et al. [23]

The Hungarian SCS is an evolving system, which is characterised by considerable backwardness and delayed development, compared to Western-European school boarding solutions. The national averages hide important regional differences. According to the Canteen Panoramas [22-24], the school snack bars and vending machines are significant competitors for the SCS, and have an improving product portfolio. At the same time the SCS is not capable of meeting changing demands. A good indicator of this is the rapidly decreasing take up rate as children’s age increases. Another indicator of problems is the fact that according to the school canteen survey (Bakacs et al. [24]), 85% of parents prepared some kind of prepacked food for their children in 2013. Prepacked food was mainly sandwiches, with vegetables (41%) or without vegetables (37%), and refreshing soft drinks (50%).

Generally speaking, the Hungarian SCS faces substantial long-standing, unsolved problems, which can be attributed to the lack of monetary resources and the lack of attention from responsible organs. This motivated the government to take action to update the nutrition requirements of the public catering service, including school meals, in 2011. In 2011 the Office of the Hungarian Chief Medical Officer issued the ‘Recommendation for Public Caterers’ with nutritional standards [25]. This recommendation provided a checklist enabling to monitor the adherence to the recommendations. This document was the basis of the 37/2014 decree of Ministry of Human Capacities (MHC) on public catering [26].

The aim of the school meal provision was to reduce the prevalence of obesity and non-communicable diseases (NCDs) among Hungarian children and adolescents, as well as promote healthier environments, especially in schools. The most important elements of the decree are summarised in Table 3. This rather ambitious regulation has tried to increase the fruit, vegetable, cereals and milk consumption and decrease the consumption of fat, sugar and salt. When comparing the content of the Decree with dietary guidelines of other member states of the EU, our nation's dietary guidelines seem to be in line with the nutrition policy of most EU member states. However, the public acceptance of this new regulation has been mixed, and mainly negative. Overall take up rates were generally low, according to the comments of school children. Pupils refused the dishes which conformed to the requirements and both children and their parents rebelled against the rules. In 2015 the Hungarian Association of Dietitians carried out a survey among dietitian food-service managers about the practical feasibility of Hungarian Regulation No. 37/2014 on nutrition requirements in the provision of public food services. Of the 56 food-service managers interviewed 19 represented child nutrition institutions. Since the introduction of the regulation in 36 of 56 institutions interviewed satisfaction with nutrition care had decreased. In 13 cases the rate of dissatisfaction was 30% or more, and the amount of daily food waste increased significantly. The majority of catering service providers (62%) requested some alterations to the regulations because the prescribed composition of the food was not in line with children’s demands. The greatest cause of dissatisfaction among parents and children derived from the control of salt content, and the attempt to provide the prescribed quantity of dairy products and added sugars [27]. To date, there are no representative, academically well founded empirical studies on pupils’ consumption of the new school meals [28]. That is why, in 2016, the regulatory framework was changed [29]. The new decree significantly modified the prescriptions. The most important features of the original and the modified decree are summarised in Table 3.

Table 3. The most important changes in the public catering decrees.

2014*

2016**

Food-based   standards

Food-based   standards

              Specific foods and food groups   have to be provided daily for all age groups

                       (for one person)

5 meals/day

4 portions of fruits or vegetables per day, at least one of which   should be raw

4 portions of fruits or vegetables per day, at least one of which   should be raw

3 portions of cereals, at least one which should be whole grain

3 portions of cereals, at least one which should be whole grain

0.5 l milk or a diary product with an adequate amount of calcium

-

Nursery (1 to <3 years)

3 portions of fruits or vegetables per day, at least one of which   should be raw

3 portions of fruits or vegetables per day, at least one of which   should be raw

2 portions of cereals, at least one of which should be whole grain

2 portions of cereals, at least one of which should be whole grain

0.4 l milk or a diary product with an adequate amount of calcium

-

3 meals/day

2 portions of fruits or vegetables per day, at least one of which   should be raw

2 portions of fruits or vegetables per day, at least one of which   should be raw

2 portions of cereals, at least one of which should be whole grain

2 portions of cereals, at least one of which should be whole grain

0.3 l milk or a diary product with an adequate amount of calcium

-

1 meals/day

1 portion of fruits or vegetables per day, at least three of which   should be raw over a 10-day catering period

1 portion of fruits or vegetables per day, at least three of which   should be raw over a 10-day catering period

Supplementation

If pre-primary, 5 or 3 meals are provided a day, milk or a diary   product with an adequate amount of calcium should be served every day

Regulations,   limitations and prohibitions of using certain foods and products

Fat content of milk

-2.8% or 3.6% milk fat milks should be served for age group 1-3

-1.5% or <1.5% milk fat milks should be served above 3 years old

-2.8% or 3.6% milk fat milks should be served for age group 1-3

-2.8% or <2.8% milk fat milks should be served above 3 years old

Water

Constant access to fresh water (outside of bathrooms)

Constant access to fresh water (outside of bathrooms)

Added sugar

Free sugar should not exceed 8% of total energy in a 10-day catering   period

Free sugar should not exceed 10% of total energy in a 10-day catering   period

Salt and free sugar

Salt and sugar should not be placed on dining table.

Salt or sugar storers should be labelled: "Excessive salt intake   could cause cardiovascular diseases, obesity and diabetes! "

*Public Catering Decree EMMI (Ministry of Human Capacities), Decree 37/2014 (came into force in September 2015)

** Decree on modification of EMMI (Ministry of Human Capacities), Decree 37/2014 (came into force in December 2016)

Source: MHC [26] and MHC [29]

The Hungarian SCS is a highly complex, dynamic system. The last Decree on Public Catering has yielded important results: the quality of meals has been increasing in the last few years [30]; however, numerous unresolved problems have remained. The new regulatory framework of the SCS is more liberal, although in numerous points it represents a step back from the goals of the former Decree. The aim of this study is to uncover the causes of this phenomenon. In a wider context our goal is to understand the justification of the low acceptance of the SCS towards a regulatory attempt to manage a long-standing, generally accepted problem, namely childhood obesity. What is the main reason that the regulation could not be put into practice? How can we explain that the take up rate of the SCS has not increased considerably since the introduction of the new Decree. To achieve this goal the authors have applied an innovative, semi quantitative method for the analysis of social bargaining.

Provide a clearer introduction to MACTOR The MACTOR analysis seems to be very interesting, but is likely many readers will be unfamiliar with it. Please include a clear, succinct explanation of what it is. Are all of the following terms part of a MACTOR analysis, what does each mean (e.g., direct influences, interest relations, influence/dependence, mobilizing force, and convergence of actors)? Please use the same term in the methods as you do in the rest of the article – consistency helps with clarity

Response

The fundamental theoretical paradigms of the analysis were institutional economic theory [31, 32], principle-agent theory [33] and the concept of strategic planning [34]. According to the basic theory of the so-called “French school of strategy” the different social systems can be considered as a playground in which different groups of participants (the actors) take part with the purpose of making their specific interests prevail. In the opinion of [35], if one can adequately simplify the actors and the most characteristic features of their systems of interests and strategies, then it is possible to analyse the chances of different actors realizing their goals. The method of the systematic analysis of social bargaining can be described by using the MACTOR model. One of the key concepts of the model is that actors may influence other actors in terms of their potential to put pressure on other actors directly or indirectly in order to affect their behaviour. The influence of one actor (A) on another actor (C) is the sum of the direct and indirect influences of actor A on actor C.

Based on unstructured interviews, the key actors of the catering system were determined. In the next phase the intensity of mutual direct influences was characterized using a rectangular matrix offering a good overview of the MACTOR method. The cells of the matrix – by definition – reflect the intensity of the influence of any actor in a row on any actor in a column. The intensity of the direct influence by one actor on another was measured on a 0-4 scale ranging from no influence to absolute influence.

The importance of different goals from the point of view of each actor was expressed by the Matrix of Actor-Objective. This was the so-called 1MAO matrix. Each cell of the matrix contained the attitude of a given actor towards a given goal in the form of a positive, 0 or negative sign. In the second phase the 2MAO matrix is determined, which contains the intensity of these attitudes determined from the point of view of different actors and quantified on a -4 …+4 scale, where -4 denotes the high importance and total negation of the given goal, and +4 denotes the high importance and total support.

The mathematical methodology of the MACTOR method is presented in the literature [36].

Explain clearly how you selected participants, especially teachers, parents, and local authorities and summarize the interview questions In this section, please explain information as clearly as possible. As a small example, on line 122 you indicate you interviewed over 20 stakeholders and on line 132 that you interviewed 24 stakeholders. If this group is the same, then on line 122 simply indicate you interviewed 24 stakeholders. Later, on line 175, it indicates that you conducted 18 interviews and that Table 1 contains the structure of respondents, however, Table 1 is the 24 participants from the preliminary phase, which is confusing. Please clarify. How did you choose parents, teachers, and local authorities? Did you interview people from the education and health side of government or other government agencies, such as agriculture? Because they can have different views, do you think it’s appropriate to label them only as ‘government?’ How do you account for the 84 people listed in table 2 when you indicated you interviewed 33 experts but provide no further information about how you selected additional participants? What kinds of questions did you ask participants – did you ask all of them the same types of questions or did it vary by their roles? This information is vital! (The supplementary information provides little information, I suggest deleting it).

Response

The data collection for the analysis was a multiphase process.

In the present study we applied a self-designed interview method. Besides analysis of publicly available papers, press releases, newspaper articles and the blogosphere, face-to-face expert estimations were made with 24 stakeholders related to the field of Hungarian catering. This series of preliminary interviews were conducted with the purpose of determining the set of relevant actors and interests. The interviews were carried out in 2015 and 2016. The aim of this preliminary phase of interviews was to outline the most important stakeholder groups and the set of the potential objectives of the stakeholders. As a result of these preliminary investigations a robust and relatively well-manageable set of actors and goals could be identified. In setting up a pool of experts a specific procedure was followed. In this phase we pursued the following logic. We considered experts to be people (1) who have a direct “field” experiences in catering functions as parents or teachers; (2) people whose job directly involves a catering business with relatively long experience in the practice of SC and whose existence directly depends on this enterprise S; (3) independent experts, preferably those who have been especially active in professional social debates concerning the catering regulations in the printed and electronic media; (4) experts who have been actively involved in the preparation and enforcement of the new regulatory framework of the SCS. The attitude of experts towards school catering has not been taken into consideration, neither in the choice of experts, nor in the interview phase.

The second phase of research was a semi-quantitative interview. The list of potential participants was collected on the basis of intensive research into publications (including professional conferences, various formal and informal meetings of professional communities, the blogosphere and the grey literature), membership of professional organizations and the personal recommendations of other experts.

In summary, the names of 321 experts were collected (not including parents and teachers). Out of these experts we tried to make contact with specialists who supposedly – in the opinion of at least two members of the community of authors – have a more ‘holistic’ approach to the SCS universe without taking into consideration their attitude to the SCS question. In this way 78 experts were selected. We contacted 61 of them; 45 respondents expressed their willingness to participate in the research. Due to time and financial constraints 33 expert-interviews were carried out, all of them face-to-face. Additionally, we interviewed 26 parents and 13 teachers. The characteristic features of interviewees re summarised in Supplement 1.

The quantification of the intensity of actor-actor influences, as well as the actor-objective relations has been developed in a step-by-step manner. As we have experienced with our previous research [37], filling out the input matrices in the form of MS Excel worksheets for research was a very time-consuming (and in some cases a rather boring) process often leading to internal contradictions because it was very difficult to achieve a general common interpretation of different scales. That is why a semi-structured interview was used [38]. The conversion of the verbal estimations was carried out in the personal interview phase with the help of the researchers. The only task of the researchers was to help to interpret the different scales. This method provided to be an efficient method for achieving internal consistency in the input data for analysis [39].

In the framework of the interviews we asked the respondents to evaluate the bargaining power of each actor in comparison with another actor (e.g. Government vs. teachers, Government vs. Catering service managers etc…) on a 0-4 scale. The interpretation of this scale was the following:

0 – no direct influence

1 – actor A can eliminate the tactical steps of actor B

2 – actor A can jeopardize/eliminate the projects of actor B

3 – actor A can jeopardize/eliminate the strategic goals of actor B

4 – actor A can substantially influence/dominate actor B

In the second phase of interviews we asked the interviewees to evaluate the attitudes of actors (stakeholders) towards different elements of goal set on a -4… 0 … +4 scale. The interpretation of the scale was as follows:

-4 the objective is against the vital interest/jeopardizes the existence of the actor

-3 the objective jeopardizes the strategic mission of actors

-2 the objective jeopardizes the tactical goals of the actors

-1 the objective jeopardizes the operative goals of the actor

0 the actors’ attitude towards the goal is neutral

+1 the objective falls in line with the operative goals of the actor

-2 the objective falls in line with the tactical goals of the actors

3 the objective considerably supports the strategic goals of the actor

4 the objective is a vital interest of the actor

The participants received the cumulated input–matrices and their interpretation by e-mail, and had the opportunity to suggest some modifications. The results of the MACTOR analysis were discussed in detail with a representative pool of respondents in a group discussion and in face-to-face interviews. This phase of the research was an explorative one since our ambition was not to create a representative sample but rather to collect a relatively wide range of opinions.

Figure 1. Flowchart of investigations.

Source: Authors’ own construction

The research topics were considered very important and interesting questions by all participants, which is why the willingness to participate in the interviews was very high. People evaluated their participation in the research very positively and they were cooperative. Participants were willing to share their experiences and views with others on different problems related to the school catering system development.

It was a very frequent reflection that: ”There is so much talking about school feeding but this is the first time that my opinion is important for someone”. Or, as one parent formulated: “This is a very long-lasting problem and a crucial issue but lack of money and attention is an obstacle to improve the situation, so there is not too much to expect”.

4. Shorten the ethics section Basically, all we need to know is that the research was approved through an ethics review, it doesn’t need a separate section.

Response

Thank you. Correct. Ethics section has been shortened: The Inter-Faculty Research Ethics Committee of the faculty of Budapest Corvinus University approved both the concept and procedure of the research (Ref. No.15/12/2014). All of the participants signed an informed consent, which described the procedure of research in details

Provide the results you indicate you are going to provide. This section needs significant improvement On line 207, you indicate the stakeholders provided opinions and identified key goals, then that’s what this section should report. Here is some more specific feedback to assist you:

A. In the “Government” sub-section, you indicate the policy was not a priority and its implementation was complicated by government bureaucracy. After that, it’s not clear if the next part of that section reflects any opinions or key goals. Information on local products and other information is included that needs to be referenced. To my earlier point, were there different views among different stakeholders from the same group, such as government?

Response

Analysing the map of influences and dependences between actors (Figure 2) it is obvious that the government has a relatively favourable bargaining position because it has a relatively high level of influence and a low level of dependence. The direct socio-economic environment of children’s food consumption is the following: the triangle of teachers, local authorities and parents have roughly the same position, namely a relatively high level of influence and a low level of dependence. The owners of the SCS firms have approximately the same level of dependence as the former three actors, with a much lower influence. The two key actors of the SCS system, the children and the catering service managers, have an extremely low level of influence, which – especially in the case of the children – is accompanied by high dependence. In other words, the two critical actors of the systems, namely the actual service providers and the children have the least possibility to influence the operation of the system.

Figure 2. The influence-dependence relations of actors.

Source: Authors’ own construction

B. For most of the next sub-sections, Local Authorities, Catering Service Providers, etc., it doesn’t appear that any opinions or key goals are reported. It isn’t clear why you chose to include the information you did.

Response

You are right. This subsection has been rephrased.

C. The sub-section on the children isn’t about the children, it’s about the environments and employees (you didn’t interview children – why have a section on them?).

Response

Correct. We did not interview children due to the difficult ethical regulatory framework but we are convinced that based on the information provided by teachers, parents and the school catering specialists we are able to formulate some knowledge on the characteristic features of the behaviour of childrens.

D. Why are there no actual quotes from participants in this article? It seems disconnected from real people. E. It is difficult to interpret the results without a clearer methods section, such as how is the Mobilizing Force calculated and what does it mean, and likewise there is a very limited description of Figure 2, so, because I don’t really understand the results as presented, I have fewer comments on this section.

Response

Thank you. Just a few example:

A catering manager said: “I am sick and tired to hear from the early morning to the late night each day the sniveling spoiled children. They encourage each other to refuse to eat our food, and they are complaining about the bad meal we prepare.”

The catering managers are frontline solders of the system, however, they felt themselves abandoned. They were complaining the following way: “from this limited material resources (lack of money) we are not able to buy the raw materials for such food, which could satisfy the requirements of Decree… I do not know enough recipes to prepare a various (diverse) menu satisfying the requirements – we did not receive any help to do this.”

Our interviews highlighted that in numerous cases the parents do not have enough time to cook, even they often go to fast food restaurants. This phenomenon decreases families’ influence on the healthy nutrition of children and adolescents. All of the respondents agreed, that due to considerable differences in socio-economic structure in Hungary, it is hard to speak about parents, as a homogenous group, because (1) there are parents, who simply do not have energy/time to take care with food consumption of their children. (2) who are worrying about the low quality/quantity of food, served to their children in school canteens, that’s why they pack them sandwiches or give them money to buy additional food, (3) in case of poor families the school canteen plays an important role in disburdening of families (and their budget) from daily food provision. One school canteen manager has told: “the most important days for us are the Mondays and the Fridays: On Monday we have to prepare to energy-rich food, because the children wants nutrient rich food after the unsatisfactory food consumption in the week and, and on Friday we have to “fill up” the children with copious food”.

Our interviews highlighted that the current world of the SCS is quite distant from the demands of children. There is an old, adolescent-sized furniture and sometimes rude and un-motivated kitchen staff (consisting mainly of older, often burned-out female employees), who acquired their experiences in years of relative food shortage and are not able to communicate in an appropriate way with children. Catering reform will only be successful if pupils like and choose to eat these meals, but this aspect has been neglected in Hungary.

The majority of specialists agreed that time has passed the old-style catering infrastructure by, and it has not been renovated for decades. This contrasts noticeably with the vivid colours and modern interiors of the majority of fast-food restaurants which target the younger generation. Under these conditions there is just a relatively low chance of attracting young consumers, who often consider traditional food as old-style.

Our interview partners agreed that theoretically teachers play a very important role in the formation of the eating behaviours and eating habits of children. The teachers eat mainly the same food of the children, if they are eating in the same canteen.

Under these conditions there is an increasing tendency for overburdened, burnt out teachers. As one teachers formulated it: ”I have a lot of problems in school and in private life, I am simply too tired to deal with such problems, as the nutrition of the children”. As another has formulated: “I am fed up with the fact that the society tries to push all the problems to the schools and teachers. I do not feel it to be my responsibility to care with eating of the children under such conditions, when I have to teach them the most elementary rules of social behaviour, just because their parents are playing with their smartphones on lingering social websites?”.

The indicator “mobilizing force” of different goals has been calculated based on acceptance of differing goals weighted by bargaining power (influence) on actors.

Analysing the mobilizing force of the different interests good taste and children’s health have the highest value (Table 6). Notwithstanding, it is important to stress that the mobilizing force of the sum of cost minimization and simplicity is much higher than the healthiness of food criteria.

Table 6. Mobilising force of different goals.

Goal

Mobilising force

good taste of the meal

12.8

healthy food

17.1

healthy children

17.8

vote maximisation

8.6

feeling to be sated

10.2

minimisation of expenditure on health   promotion

17.4

simplicity of food preparation

2.4

Source: Authors’ own construction

 Strengthen the discussion – base it on results/data and information about school nutrition policy On line 402, you indicate the take up was very low but the assessment from 2017 (articles I indicated) appear to conflict with that finding – this difference needs to be resolved. The title of the article may need revising. On line 130 you indicated you analyzed available papers, press releases, etc., and on line 406 you indicated that negative opinions were reinforced by the media, but you provided no data or results in the results section of the article. On line 409 you mention about modifying the legislation. Did this happen and if so, what changes were made? If it did not happen, then this line needs to be re-written to be clearer. Again in this section, reference all external information, such as the WHO twelve steps to healthy eating. The last three references don’t seem to make sense. There is lots of international literature on school food and nutrition policy development, implementation, and monitoring. The article will be stronger if you stay within that topic.

Response

Thank you for your comments. Generally speaking, the Hungarian SCS faces substantial long-standing, unsolved problems, which can be attributed to the lack of monetary resources and the lack of attention from responsible organs. This motivated the government to take action to update the nutrition requirements of the public catering service, including school meals, in 2011. In 2011 the Office of the Hungarian Chief Medical Officer issued the ‘Recommendation for Public Caterers’ with nutritional standards [25]. This recommendation provided a checklist enabling to monitor the adherence to the recommendations. This document was the basis of the 37/2014 decree of Ministry of Human Capacities (MHC) on public catering [26].

The aim of the school meal provision was to reduce the prevalence of obesity and non-communicable diseases (NCDs) among Hungarian children and adolescents, as well as promote healthier environments, especially in schools. The most important elements of the decree are summarised in Table 3. This rather ambitious regulation has tried to increase the fruit, vegetable, cereals and milk consumption and decrease the consumption of fat, sugar and salt. When comparing the content of the Decree with dietary guidelines of other member states of the EU, our nation's dietary guidelines seem to be in line with the nutrition policy of most EU member states. However, the public acceptance of this new regulation has been mixed, and mainly negative. Overall take up rates were generally low, according to the comments of school children. Pupils refused the dishes which conformed to the requirements and both children and their parents rebelled against the rules. In 2015 the Hungarian Association of Dietitians carried out a survey among dietitian food-service managers about the practical feasibility of Hungarian Regulation No. 37/2014 on nutrition requirements in the provision of public food services. Of the 56 food-service managers interviewed 19 represented child nutrition institutions. Since the introduction of the regulation in 36 of 56 institutions interviewed satisfaction with nutrition care had decreased. In 13 cases the rate of dissatisfaction was 30% or more, and the amount of daily food waste increased significantly. The majority of catering service providers (62%) requested some alterations to the regulations because the prescribed composition of the food was not in line with children’s demands. The greatest cause of dissatisfaction among parents and children derived from the control of salt content, and the attempt to provide the prescribed quantity of dairy products and added sugars [27]. To date, there are no representative, academically well founded empirical studies on pupils’ consumption of the new school meals [28]. That is why, in 2016, the regulatory framework was changed [29]. The new decree significantly modified the prescriptions. The most important features of the original and the modified decree are summarised in Table 3.

Table 3. The most important changes in the public catering decrees.

2014*

2016**

Food-based   standards

Food-based   standards

              Specific foods and food groups   have to be provided daily for all age groups

                       (for one person)

5 meals/day

4 portions of fruits or vegetables per day, at least one of which   should be raw

4 portions of fruits or vegetables per day, at least one of which   should be raw

3 portions of cereals, at least one which should be whole grain

3 portions of cereals, at least one which should be whole grain

0.5 l milk or a diary product with an adequate amount of calcium

-

Nursery (1 to <3 years)

3 portions of fruits or vegetables per day, at least one of which   should be raw

3 portions of fruits or vegetables per day, at least one of which   should be raw

2 portions of cereals, at least one of which should be whole grain

2 portions of cereals, at least one of which should be whole grain

0.4 l milk or a diary product with an adequate amount of calcium

-

3 meals/day

2 portions of fruits or vegetables per day, at least one of which   should be raw

2 portions of fruits or vegetables per day, at least one of which   should be raw

2 portions of cereals, at least one of which should be whole grain

2 portions of cereals, at least one of which should be whole grain

0.3 l milk or a diary product with an adequate amount of calcium

-

1 meals/day

1 portion of fruits or vegetables per day, at least three of which   should be raw over a 10-day catering period

1 portion of fruits or vegetables per day, at least three of which   should be raw over a 10-day catering period

Supplementation

If pre-primary, 5 or 3 meals are provided a day, milk or a diary   product with an adequate amount of calcium should be served every day

Regulations,   limitations and prohibitions of using certain foods and products

Fat content of milk

-2.8% or 3.6% milk fat milks should be served for age group 1-3

-1.5% or <1.5% milk fat milks should be served above 3 years old

-2.8% or 3.6% milk fat milks should be served for age group 1-3

-2.8% or <2.8% milk fat milks should be served above 3 years old

Water

Constant access to fresh water (outside of bathrooms)

Constant access to fresh water (outside of bathrooms)

Added sugar

Free sugar should not exceed 8% of total energy in a 10-day catering   period

Free sugar should not exceed 10% of total energy in a 10-day catering   period

Salt and free sugar

Salt and sugar should not be placed on dining table.

Salt or sugar storers should be labelled: "Excessive salt intake   could cause cardiovascular diseases, obesity and diabetes! "

*Public Catering Decree EMMI (Ministry of Human Capacities), Decree 37/2014 (came into force in September 2015)

** Decree on modification of EMMI (Ministry of Human Capacities), Decree 37/2014 (came into force in December 2016)

Source: MHC [26] and MHC [29]

The Hungarian SCS is a highly complex, dynamic system. The last Decree on Public Catering has yielded important results: the quality of meals has been increasing in the last few years [30]; however, numerous unresolved problems have remained. The new regulatory framework of the SCS is more liberal, although in numerous points it represents a step back from the goals of the former Decree. The aim of this study is to uncover the causes of this phenomenon. In a wider context our goal is to understand the justification of the low acceptance of the SCS towards a regulatory attempt to manage a long-standing, generally accepted problem, namely childhood obesity. What is the main reason that the regulation could not be put into practice? How can we explain that the take up rate of the SCS has not increased considerably since the introduction of the new Decree. To achieve this goal the authors have applied an innovative, semi quantitative method for the analysis of social bargaining.

The article requires a thorough edit for readability, use of language (e.g., is this group capable of indicating employees are burned out?), and spelling (e.g., line 293 hardy should be hardly).

Response

Correct. Thank you. Language and grammar have been edited under the guidance of a professional native speaker.

Round 2

Reviewer 2 Report

Reviewer Comments

Nutrients: The Reform of School Catering In Hungary: Anatomy of a Health Education Attempt

Summary:

I applaud the authors for the considerable effort that they have put into revising the paper. It is much improved. However, some concerns remain and a careful edit by another native English speaker would be helpful before being accepted for publication.

Major Issues

Introduction:

Please add which standards were used to quantify prevalence of child overweight and obesity in Hungary.

Please add information on Hungary’s SCS to Table 1 as well as whether meals are offered free, reduced price or full price to students.

In Table 2, please add footnotes to explain what is meant by boarding, and energy and nutrient content calculation, and explain whether foods sold in the snack bar and vending machine are outside of the rules of the SCS or have other requirements.

In Table 3, please clarify what ages the 5, 3 and 1 meal/day requirements apply to. You may want to add a heading to signify the ages and put the standards for the youngest children at the end. If it does not take too much space include a footnote to explain the rationale for the raw fruit/vegetable requirement, and to explain what ‘constant’ water means. Besides the salt shaker on the table, what were the other sodium restrictions? Given that the aim was to reduce obesity, were there any limits on the total calories of meals?

It would be helpful to describe what monitoring and enforcement of the school meal standards is done in Hungary, how often and who is responsible for this monitoring and enforcement.

Methods:

Please justify, given your conclusion that children were not consulted during the process of the school meal reform, why you did not include students in your evaluation/interviews.

Please clarify what is meant by a more ‘holistic approach to the SCS universe’ view of stakeholders.

The figure is very helpful – thanks for adding it. Please fix a few typos.

Please add how parents and teachers were identified and recruited. Were they from the same schools or geographic areas as the other stakeholders?  How did you ensure getting a variety of opinions within each stakeholder group?

Line 507-10: It seems odd to include stakeholder quotes in the methods section, especially the second quote. Consider moving to the results.

Results:

Elected officials in Hungary sometimes give Christmas gifts to their electorate? Is this common? If not, consider selecting a more common example or delete this example.

Line 661-664: Was the information from an anonymous source one of the stakeholders interviewed?  If not, please delete this comment or move to the discussion section and only include if verified by other sources. If yes, please explain what the intent is of ‘spying’ on the caterers.

Table 4. Please add what a 4 means on the scale of 0 to 4. Does it mean complete control or strong influence, for example?

Line 743: This seems like a contradiction: “a relative majority of a small group’.

Table 5: Please add what a score of -4 vs 0 vs +4 means. How were the goals in the table derived?

Line 759: Statements like ‘it was surprising to see how important school feeding was for parents’ and that this ‘can be considered favorable’ need to go in the discussion and then put in the context of why this was unexpected or favorable. What prior studies or information was the assumption to the contrary based upon? In the results section you should only say that you found that school feeding was important to parents.

Table 6: Please add how the values in the table were calculated and what a high versus low value means. Does it make sense to list these from highest to lowest?

Discussion:

Line 802 “Our result are in line with conclusions drown by another actors” Is an example of a sentence that is not clearly written. Do you mean that your results are consistent with prior studies?

Table 1 (which is the second Table 1, now oddly at the end of the paper and not referenced in the results): Round % to whole numbers and reference in the results section.

Table 2 (also needs to be referenced in the results section and renumbered so that there are not two Table 2s): Add the total sample sizes for each type of stakeholder (in each row) and for total stakeholders (in table title). Also it would be helpful to add that these are participants in the semi-structured interviews and to reference this table in the results.

Author Response

Review 2

Comment

Please add which standards were used to quantify prevalence of child overweight and obesity in Hungary.

Response

Thank you for your comments. We have inserted the following information: This causes a considerable burden for the social security and health care system [14, 13] according to the categorisation of nutritional status of people based on the body mass index (BMI) [15]. Obesity has been described by the WHO as “a global epidemic” due to its high prevalence. It is well known that there are different methods for evaluation of nutritional status of children [16]. Hungary has applied different categorisation of nutritional status of children and youth, for example percentiles [17], waist circumference [18] or BMI [19]. All of the researchers have agreed that increasing of obesity is a continuous trend in Hungary.

Comment

Please add information on Hungary’s SCS to Table 1 as well as whether meals are offered free, reduced price or full price to students.

Response

The paragrahp has been modogoied the following way: Hungary’s SCS is a subsidised system where meals are offered by a reduced price and in some cases (based on social position) the school meal is free. The Hungarian National Survey of SCS (called as Canteen Panorama) is a regular report of the National Institute of Pharmacy and Nutrition (former Hungarian Office of Food and Nutrition) [30-32]. The monitoring and enforcement of the school meal standards is a mandatory regular activity of the National Food Safety Office (NFSO) with government functions. The essential results of the last three surveys are summarised in Table 1. Vending machines and snack bars are not considered as a part of SCS, but their product portfolio is strictly regulated.

Table 1 has been removed following the comments of Review 3.

Comment

In Table 2, please add footnotes to explain what is meant by boarding, and energy and nutrient content calculation, and explain whether foods sold in the snack bar and vending machine are outside of the rules of the SCS or have other requirements.

Response

“Boarding” has been rephrased to “Schools offering warm meal at least once a day with a canteen”.  (please see Table 1). Vending machines and snack bars are not considered as a part of SCS, but their product portfolio is strictly regulated.

Footnote:

** Energy and nutrient content calculation is based on official nutrient and energy content information, the age-specific menu is calculated based on this information and in some cases even a specific menu planning sotftware is applied.

Comment

In Table 3, please clarify what ages the 5, 3 and 1 meal/day requirements apply to. You may want to add a heading to signify the ages and put the standards for the youngest children at the end. If it does not take too much space include a footnote to explain the rationale for the raw fruit/vegetable requirement, and to explain what ‘constant’ water means. Besides the salt shaker on the table, what were the other sodium restrictions? Given that the aim was to reduce obesity, were there any limits on the total calories of meals?

Response

Thanks. Correct. This Table has been modified accordingly and a new Table added. The ration of the raw fruit /vegetable requirement is not included in the legislation but it is assumed that this can be explained by higher vitamin content of these goods [38].

Comment

It would be helpful to describe what monitoring and enforcement of the school meal standards is done in Hungary, how often and who is responsible for this monitoring and enforcement.

Response

The monitoring and enforcement of the school meal standards is a mandatory regular activity of the National Food Safety Office (NFSO) with government functions. Vending machines and snack bars are not considered as a part of SCS, but their product portfolio is strictly regulated.

Comment

Methods:

Please justify, given your conclusion that children were not consulted during the process of the school meal reform, why you did not include students in your evaluation/interviews.

Response

Children were not included in our evaluation/interview because we focussed on the socio-economic arena of SCS. The primary aim of our investigation was the determination of optimal (qualitative and quantitative) parameters of school meals. 

Comment

Please clarify what is meant by a more ‘holistic approach to the SCS universe’ view of stakeholders.

Response 

In the process of our analysis we tried to take into consideration all relevant stakeholders.

Comment

The figure is very helpful – thanks for adding it. Please fix a few typos.

Response

Thanks. It has been done.

Comment

Please add how parents and teachers were identified and recruited. Were they from the same schools or geographic areas as the other stakeholders?  How did you ensure getting a variety of opinions within each stakeholder group?

Response

Specific attention was paid to choose parents and teachers from relatively well-off and less developed regions of Hungary – cities like Budapest and Szeged, small town (Hajós) and village (Báta). Personal acquaintance played an important role in the choice of teachers and parents.

Comment

Line 507-10: It seems odd to include stakeholder quotes in the methods section, especially the second quote. Consider moving to the results.

Response

Stakeholder quotes have been moved to the results section.

Results:

Comment

Elected officials in Hungary sometimes give Christmas gifts to their electorate? Is this common? If not, consider selecting a more common example or delete this example.

Response

Thanks. We have inserted the following sentence: This rather curious behaviour is an integrated part of the paternalistic political culture of Hungary [50].

Comment

Line 661-664: Was the information from an anonymous source one of the stakeholders interviewed?  If not, please delete this comment or move to the discussion section and only include if verified by other sources. If yes, please explain what the intent is of ‘spying’ on the caterers.

 Response

Correct. This sentence has been removed.

Comment

Table 4. Please add what a 4 means on the scale of 0 to 4. Does it mean complete control or strong influence, for example?

Response

We have insterted a new sentence into the heading of the table: The matrix of direct influences on actors measured on a 0-4 scale (0 – no direct influence, 4 – very strong influence)

Comment

Line 743: This seems like a contradiction: “a relative majority of a small group’.

Response

We have added a new sentence to explain the situation in Hungary: Due to lack of democratic roots and traditions it is quite difficult to achieve parents’ influence based on an absolute majority of the parents and not just on a small group with active and noisy members.

Comment

Table 5: Please add what a score of -4 vs 0 vs +4 means. How were the goals in the table derived?

Response

The goals in this table are based on preliminary interviews.

Interpretation: -4 the objective is against the vital interest/jeopardizes the existence of the actor, 4 the objective is a vital interest of the actor.

Comment

Line 759: Statements like ‘it was surprising to see how important school feeding was for parents’ and that this ‘can be considered favorable’ need to go in the discussion and then put in the context of why this was unexpected or favorable. What prior studies or information was the assumption to the contrary based upon? In the results section you should only say that you found that school feeding was important to parents.

Response

The misleading sentence has been removed and the paragraph rephrased: According to our findings school feeding was important to parents.

Comment

Table 6: Please add how the values in the table were calculated and what a high versus low value means. Does it make sense to list these from highest to lowest?

Response

Calculation of values are based on equations given by Figure 1., values in a metric, dimensionless value.

Discussion:

Comment

 Line 802 “Our result are in line with conclusions drown by another actors” Is an example of a sentence that is not clearly written. Do you mean that your results are consistent with prior studies?

 Response

Thank you. We have rephrased this sentence: Our results are consistent with….

Comment

Table 1 (which is the second Table 1, now oddly at the end of the paper and not referenced in the results): Round % to whole numbers and reference in the results section.

Response

We have re-numbered the tables and rounded the values.

Comment

Table 2 (also needs to be referenced in the results section and renumbered so that there are not two Table 2s): Add the total sample sizes for each type of stakeholder (in each row) and for total stakeholders (in table title). Also it would be helpful to add that these are participants in the semi-structured interviews and to reference this table in the results.

Response

Thanks. We have modofied Tables in the Supplement)

Author Response

Review 3

Comment

Stay focused on school food and nutrition policy 

I hope I did not mislead you with earlier comments. I think it improves the article significantly that you now include the results from the Canteen Panorama surveys and the standards from Hungary.  They provide important context and allow the reader to assess the extent of change.  That said, I suggest you delete Table 1 on the school meals and nutrition education because they are not the direct focus of this article.  In addition, the article on which the table is based is from 2008 and school food programs are dynamic and have likely changed since then.  Furthermore, in column 1 you indicate a number of countries that do not provide funding.  At least some of these countries provide funding but not for universal, free school meals’ programs.  So – bottom line is that I think it’s extra information you don’t need, it could be out of date, and as written, is not 100% accurate. 

Response

Thank you for your valuable comments. Correct. Table 1 has been deleted.

Comment

Instead, it would be helpful to elaborate a little on your sentence that tells us that all member states of the EU have policies to help schools provide healthier meals.  For example, who has developed them (e.g., Health or Education or Agriculture or . . .) are they nutrient based and/or food based, require organic and/or local food – information that helps readers place the policy from Hungary in context.  I would not spend a lot of time/space on this item.

Response

We have inserted the following subparagraph: The state of school feeding is considered as a question of strategic importance all over the world [25]. All member states of the EU have policies to help schools to provide nutritionally balanced meals [26]. Based on the EU survey [26] and the cluster analysis [27] we have constructed a general synoptic table (Supplement 1, Table 1). It is obvious that there are considerably differences in the school feeding policy of various EU member states (Figure 1). The Hungarian regulation is rather similar to the Spanish and Danish systems.

Comment

Clarify Table 2

These points are small. 

First, it is unclear what you mean by “Boarding.”  Does this mean the children stay at school overnight or does it mean that the schools are part of school boards, or does it mean something else? 

Second, please format the table so that the proportion of students eating at the school canteen in primary, etc. is clearer.  Currently, there are no values beside primary. 

Third, if there are no vending machines in elementary schools, please let us know (currently that cell is blank).

Response

“Boarding” has been rephrased to “Schools offering warm meal at least once a day with a canteen”.  (please see Table 1). Vending machines and snack bars are not considered as a part of SCS, but their product portfolio is strictly regulated.

Comment

Streamline Table 3

Table 3 contains a lot of repeated information.  If you could indicate what is the same and what is different in a more visual way, it would help.  For example, might you have one part of the table that indicates what is the same about 2014 and 2016 (one column) and then a separate sub-set of the table with two columns indicating the differences?

Response

Correct. Thank you. We have modified Table 3 accordingly.

Comment

Clarify Materials and Methods further

This section is much improved!  Figure 1 is helpful – I wonder if it would be useful to extend it to include all the analysis (such as the Mobilising Forces, convergence)?  Also, to help guide us, it would be useful if your figure and narrative text corresponded 100%.  For example, in the figure you indicate you conducted preliminary interviews with 24 people.  In the narrative, you indicate you had a discussion with a representative pool of respondents.  I’m assuming that you mean the 24 people but if you added (n=24) at some point in the narrative, then I would know for sure.  And, if I’m wrong, then somehow you need to write it so that I understand everything clearly.

Response

We have re-formulated Figure. We have rephrased the sentence in the following way: In the framework of individual comprehensive interviews with 24 experts offered a favourable opportunity for understanding the views of different expert actors and their key goals.

Comment

There are still a few gaps.  First, we still need to know how you obtained your sample of parents and teachers.  Second, we need to know the background of the Mobilising Force measure and how to interpret the results that we see later; and likewise, how is convergence determined?

The last two paragraphs on this section seem to be results not materials and methods.

Response

Specific attention was paid to choose parents and teachers from relatively well-off and less developed regions of Hungary – cities like Budapest and Szeged, small town (Hajós) and village (Báta). Personal acquaintance played an important role in the choice of teachers and parents. The characteristic features of interviewees are summarised in Supplement (Table 3). The calculation of mobilising force is included in Figure1.

Comment

The heading for Ethical issues could be changed to ethics (there weren’t any issues) - thanks for shortening that section!

Response

Thanks. This heading has been changed to Ethics.       

Comment

Fully explain each result

For the most part, the results are explained and I think it strengthens the article to add quotes (although interesting that you don’t indicate the source of the quote, such as teacher, caterer, parent).  I still have some questions. 

First, in Table 4, what do the relatively high number of 0’s and 1’s mean – for example, that there is little shared influence within the group, or that school food and nutrition policy is a relatively low priority, or something else?

Response

The analysis of influence dependence matrix offers information on estimated power relationships between actors and also on the way of thinking of respondents. It can be considered as a rather negative tencency that interviewees evaluated the catering system as a set of different, relavitevly separated actors instead of a coherent system with numerous relations. This fact is experessed in a high number of zeroes indicating no infuence.

Comment

Second, what does Table 6 mean – also, although you refer to cost minimalization in the narrative text, it is not in the list.  I suggest for Table 6 that you list the results in ascending or descending order to make it easier for the reader to follow.

Response

The sentence has been rephrased: Notwithstanding, it is important to stress that the mobilizing force of the cost minimization of health expenditure promotion and simplicity is much higher than the healthiness of food criteria.

Comment

Third, what does Figure 3 mean?  You mention convergence – does the mean convergence in the middle of the square?  Why are there 4 quadrants – do they represent something?  We need more information to understand this result.

Response

You are right. This figure has been removed.

Comment

Tie the discussion and conclusion more closely to the results

This section is the one where I suggest you take more time to reflect on what the purpose of the article is, what you found, and what it all means.  As it is now, a fair bit of the discussion is not anchored by results.  Yet, your results uncovered some very interesting tensions among the stakeholder groups involved in school food and nutrition policy. For example, you indicate that parents, catering service providers and managers, teachers, and local authorities chose to force the government to revise the standards. What if you take a look at that statement in light of your results.  As you indicate, parents in your sample were all very supportive of healthy food (another reason it’s important in your methods to indicate the source of your parent sample – maybe they were skewed toward favouring health).  You’ve got teachers who think health is important but don’t appear to make the connection between overall health and healthy eating by students.  You’ve indicated that caterers have relatively high dependence and relatively low influence, yet they are part of the group that caused the government to change. You will see that I’m not providing answers here but I am encouraging you strongly to appreciate the data you obtained and discuss it fully.  Maybe you won’t arrive at full answers either but you have the opportunity to help us better understand the tensions, at least in Hungary. 

Response

A new paragraph has been inserted: The results of this survey highlight that government support has been rather weak so far because the majority of parents accept the importance of healthy eating but they avoid the conflicts with their children. The overburdened teachers have done little to change the eating habits of the students. The caterers and the catering service managers support the best solution for themselves. At the same time the direct influence of catering managers is relatively low, however, their behaviour and attitude are of great interest in any reform. It is a contradiction that they have not been prepared to make use of these reforms. Neither the government, nor the local municipalities have been determined enough to mobilise additional financial and intellectual sources to change the situation and consequently follow the policy.

Comment

Because your analytical approach is innovative, I think it warrants some discussion by you about how it contributed to obtaining an improved understanding of the various stakeholders and their roles. 

Response

In the discussion section the following sentence has been inserted: In general it can be stated that the MACTOR method has been an efficient tool to uncover the direct and indirect force relations and motivations of different actors.

Comment

When you make the case at the end for policy, you may also add (or not) that the study of policy itself can be an educational tool (more information in this 2016 chapter from the book: Learning, Food, and Sustainability pp 201-220:  School Food and Nutrition Policies as Tools for Learning.

Overall, while my other comments will strengthen the paper, this one is the most significant – it will really help make the key difference in the quality of the paper.  Sometimes it takes an outsider to help you see what you’ve actually done! 

Response

The study of policy itself can be an educational tool. Teaching about food and nutrition policies in schools would make students active participants throughout the policy process [75].

Comment

Ask for a further edit for grammar and writing

There is still some editing to do on this article regarding word choice and spelling.  I could provide some specific examples, but I’m assuming the editor will catch them.

Response

Language and grammar have been edited under the guidance of a professional native speaker.
